# Targeted sensors for glutamatergic neurotransmission

Yuchen Hao[1,2], Estelle Toulmé[3], Benjamin König[1,2], Christian Rosenmund[3,4], Andrew JR Plested[1,2,4]*

[1]Institute of Biology, Cellular Biophysics, Humboldt-Universität zu Berlin, Berlin, Germany; [2]Leibniz-Forschungsinstitut für Molekulare Pharmakologie, Berlin, Germany; [3]Institute for Neurophysiology, Charité - Universitätsmedizin Berlin, Berlin, Germany; [4]NeuroCure Cluster of Excellence, Berlin, Germany

**Abstract** Optical report of neurotransmitter release allows visualisation of excitatory synaptic transmission. Sensitive genetically-encoded fluorescent glutamate reporters operating with a range of affinities and emission wavelengths are available. However, without targeting to synapses, the specificity of the fluorescent signal is uncertain, compared to sensors directed at vesicles or other synaptic markers. We fused the state-of-the-art reporter iGluSnFR to glutamate receptor auxiliary proteins in order to target it to postsynaptic sites. Chimeras of Stargazin and gamma-8 that we named SnFR-γ2 and SnFR-γ8, were enriched at synapses, retained function and reported spontaneous glutamate release in rat hippocampal cells, with apparently diffraction-limited spatial precision. In autaptic mouse neurons cultured on astrocytic microislands, evoked neurotransmitter release could be quantitatively detected at tens of synapses in a field of view whilst evoked currents were recorded simultaneously. These experiments revealed a specific postsynaptic deficit from Stargazin overexpression, resulting in synapses with normal neurotransmitter release but without postsynaptic responses. This defect was reverted by delaying overexpression. By working at different calcium concentrations, we determined that SnFR-γ2 is a linear reporter of the global quantal parameters and short-term synaptic plasticity, whereas iGluSnFR is not. On average, half of iGluSnFR regions of interest (ROIs) showing evoked fluorescence changes had intense rundown, whereas less than 5% of SnFR-γ2 ROIs did. We provide an open-source analysis suite for extracting quantal parameters including release probability from fluorescence time series of individual and grouped synaptic responses. Taken together, postsynaptic targeting improves several properties of iGluSnFR and further demonstrates the importance of subcellular targeting for optogenetic actuators and reporters.

*For correspondence: andrew.plested@hu-berlin.de

**Competing interest:** The authors declare that no competing interests exist.

## Editor's evaluation

This manuscript addresses the potential value of a "tagged" version of iGluSnFRs with the idea that this approach provides a more localized measure of glutamate release at synapses. Although the new sensor does not have an increase in signal-to-noise ratio, the authors nicely address the potential advantages and limitations of their sensor and the experiments provide an important test of the localized expression of such a sensor.

## Introduction

Synapses pass information from one neuron to another, but their performance is idiosyncratic and unreliable. It was long recognised from electrophysiological work that isolating single synapses reveals variability that is otherwise lost when population responses are measured. In electrophysiology,

separating out individual synaptic responses can be achieved either by reducing release probability (*del Castillo and Katz, 1954*), by blocking all other responses (*McAllister and Stevens, 2000*), using minimal stimulation (*Isaac et al., 1996*), or by examining connected pairs of neurons (*Vyleta and Jonas, 2014*) with time resolution in the ms range. In contrast, optical report of neurotransmission via membrane dyes (*Griesinger et al., 2005*), calcium dyes (*Oertner et al., 2002*; *Enoki et al., 2009*), voltage-sensing dyes (*Popovic et al., 2015*), or neurotransmitter-binding reporters offers direct spatial resolution of synaptic events, usually with a time resolution about two orders of magnitude slower. Quantitation and analysis of the relevant optical signals remain challenging (*Helassa et al., 2018*; *James et al., 2019*; *Soares et al., 2019*; *Tagliatti et al., 2020*).

Synapses are diverse in their composition and size (*Jontes and Smith, 2000*), and it is assumed that these architectural features correspond to functional differences (*Cizeron et al., 2020*). In principle, accessing individual synaptic inputs to a neuron in space and time should allow correlation of molecular architecture with functional synapse properties and thus reveal synaptic diversity, but functional properties might be less diverse than synapse composition itself (*Farsi et al., 2021*). Plastic changes in synaptic responses, generated by sensory input (*Gambino et al., 2014*), are widely taken to represent learning and memory mechanisms, and might in turn increase diversity. However, synaptic contributions to sensory selectivity seem to depend most on the number of synapses that are activated (*Scholl et al., 2020*) suggesting that synaptic diversity may not serve immediate functional purposes. Instead, because synapses use small numbers of molecules, and tend to operate randomly (*Ribrault et al., 2011*), diversity may be needed to provide robustness through redundancy. The lack of reliability of certain synapses may be key to their computational power and role in decision making (*Evans et al., 2018*). These observations strongly motivate us to develop and test optical tools that can report activity with individual synapse resolution, and with a dynamic range sufficient to discern different modes of release, and failures to release.

Early generations of genetically encoded reporters of glutamate release relied on FRET (*Okumoto et al., 2005*; *Namiki et al., 2007*). The advent of bright, single-wavelength genetically encoded reporters that directly bind neurotransmitters, such as iGluSnFR, have expanded the scope of optical report of synaptic transmission by speeding up report to the limit of optical microscopy (*Marvin et al., 2013*). This class of sensor has also been adapted to fold more quickly (using superfolder Green Fluorescent Protein (GFP)), and offer affinity and colour variants (*Marvin et al., 2018*; *Helassa et al., 2018*). For the purpose of subcellular targeting, the compact sensor architecture of iGluSnFR is easy to transplant into other host proteins.

Fusion of fluorescent probes to proteins with selective, organellar targeting allows their subcellular expression. For example, targeting of probes to synaptic vesicles (such as Synaptophluorin; *Miesenböck et al., 1998*) has enabled quantitation of vesicle fusion and recovery with high specificity (*Balaji and Ryan, 2007*; *Chanaday and Kavalali, 2018*) albeit with slow off-kinetics (*Sankaranarayanan et al., 2000*) compared to vesicle fusion. On the other hand, most genetically-encoded sensors of neurotransmitters themselves lack targeting to synaptic sites. Even when signals from the latter are localised to putative dendritic spines, the unknown proximity of the sensor to the release site introduces unwanted variability (*Soares et al., 2019*), because the glutamate signal diverges rapidly in space and time due to diffusion (*Raghavachari and Lisman, 2004*). Where spatial confinement of the reporter was achieved by exploiting targeting (*Kim et al., 2020*) or anatomy (*Helassa et al., 2018*; *James et al., 2019*; *Duerst et al., 2020*; *Jensen et al., 2019*; *Mendonça et al., 2022*), the quantitative improvement of the fluorescent report is striking.

Here, we report that fusion of iGluSnFR to the glutamate receptor auxiliary proteins γ-2 or γ-8 produces postsynaptic-targeted sensors with improved spatial resolution. The SnFR-γ2 reporter gives a high-contrast signal through enrichment at glutamatergic synapses, and surprisingly also shows more stable activity. Spontaneous and evoked release can be resolved at up to 5 Hz, allowing dissection of the presynaptic contribution to short-term plasticity and release parameters in cultured neurons at the single synapse level.

## Results

Several groups have pursued mutagenesis of the Glt1 glutamate-binding domain to reduce the affinity of iGluSnFR for glutamate (*Helassa et al., 2018*; *Marvin et al., 2018*), with aim of improving its spatial and temporal resolution. We examined several mutants using patch-clamp fluorometry (*Figure 1—figure*

*supplement 1*). A particularly interesting mutant that has not been previously reported is the Y230F substitution that removes a hydroxyl group that coordinates glutamate in the Glt1 domain. This mutation increased $\Delta F/F$, the fractional fluorescence change upon glutamate binding (possibly by reducing fluorescence in the absence of glutamate), and sped the kinetics of the off-response (*Figure 1—figure supplement 1*). As expected, the Y230F mutation also reduced the apparent affinity for glutamate. To assess the kinetics of this mutant in synapses, we turned to autaptic microisland neuronal cultures. In this preparation, a single patch-clamp electrode can produce escaping action potentials that trigger glutamate release from synaptic terminals and record the resulting postsynaptic currents. However, we were surprised to discover that the Y230F mutant gave almost no fluorescent response in autaptic hippocampal neurons following stimulation (*Figure 1—figure supplement 2*), whereas in the same cultures, the regular iGluSnFR gave robust fluorescent responses, phase-locked to the postsynaptic currents, as did Synaptophluorin (with about 100× slower off kinetics). Paradoxically, the Y230F was expressed on the neuronal surface and responded robustly to 10 mM glutamate perfusion over the autaptic neuron (*Figure 1—figure supplement 2*). Combined with the observation that the fluorescence from unmodified iGluSnFR on average occurs over a spatial extent much larger than an individual synapse (~10 μm², see below) this observation led us to hypothesise that the low-affinity sensor was on average not sufficiently close to the synaptic glutamate signal to properly report it. Other low-affinity sensors have been reported (*Helassa et al., 2018*; *Marvin et al., 2018*) but as outlined above, their localisation in relation to postsynaptic densities (and therefore the signal they report) is uncertain.

iGluSnFR is membrane targeted through a single pass pDisplay peptide (*Marvin et al., 2013*), which includes no localisation signal to concentrate sensors at or near synapses, even if its responses themselves provide a spatial readout of glutamate release. To address this aspect, we fused the iGluSnFR extracellular domain to AMPA-type glutamate receptor in order to target it to excitatory synapses. Inline fusion at a permissive site in AMPA receptor subunits GluA1 and GluA2 extracellular region produced a sensor with a very small $\Delta F/F$ (*Figure 1—figure supplement 3*). Instead, we turned to the auxiliary proteins (TARPs) that decorate the periphery of AMPA receptors. To place the iGluSnFR sensor domain on the extracellular side of the membrane we fused a single pass transmembrane helix (from the NETO2 kainate receptor auxiliary protein; *Zhang et al., 2009* to avoid any adventitious competition against the AMPA receptor–TARP interaction) to the N-terminus of Stargazin (γ-2) or γ-8 (*Figure 1A*). These chimeric sensors had similar performance to the original iGluSnFR and associated normally with AMPA receptor complexes in HEK cells (*Figure 1*), also acting to modulate AMPA receptor gating. The two reporters had similar apparent affinity for glutamate (within a factor of 2) to iGluSnFR, quite distinct from published 'low-affinity' variants (*Helassa et al., 2018*). We named these reporters SnFR-γ2 and SnFR-γ8.

Bulk rat hippocampal cultures infected with the AAV of iGluSnFR gave wide, diffuse and, in our hands, relatively infrequent fluorescent signals (see *Figure 2* and *Video 1*). We made adeno-associated viruses (AAVs) of both SnFR-γ2 and SnFR-γ8. Hippocampal bulk cultures infected with these viruses revealed regular fluorescence spikes (typically about 1 Hz frequency, see *Figure 2* and *Video 2*). At hand-picked regions of interest (ROIs), SnFR-γ2 and SnFR-γ8 showed a more localised signal (*Figure 2*).

To address synaptic localisation quantitatively, we performed immunocytochemistry on bulk cultured hippocampal neurons (*Figure 3*). We live-labelled cells for GFP (corresponding to the cpGFP domain of iGluSnFR) to show surface localisation. Following fixing, we stained for PSD95 and MAP2 to mark postsynaptic densities and dendritic shafts, respectively. The signal for SnFR-γ2 and SnFR-γ8 was markedly more punctate than for iGluSnFR, as expected. We detected about 50% greater average intensity in PSD spots for SnFR-γ2 and SnFR-γ8 (p = 0.03 and 0.06, respectively), as compared to iGluSnFR, suggesting that SnFR-γ2 and SnFR-γ8 are weakly enriched at synapses and probably exchange with extrasynaptic sites. The PSD size was slightly reduced for SnFR-γ2 compared to iGluSnFR, but the number of PSD spots per neuron imaged was similar across all conditions.

The demonstration that SnFR-γ2 and SnFR-γ8 are enriched at synapses suggested they might show reduced spatial spread of the fluorescent glutamate report in live cells. To investigate this, we co-infected the cultures with a lentivirus for Homer-tdTomato. The complexity of the SnFR-γ2 signal between intracellular (probably endoplasmic reticulum [ER]) membranes and synaptic sites forbade a simple measure of colocalisation between Homer and SnFR-γ2 in these experiments.

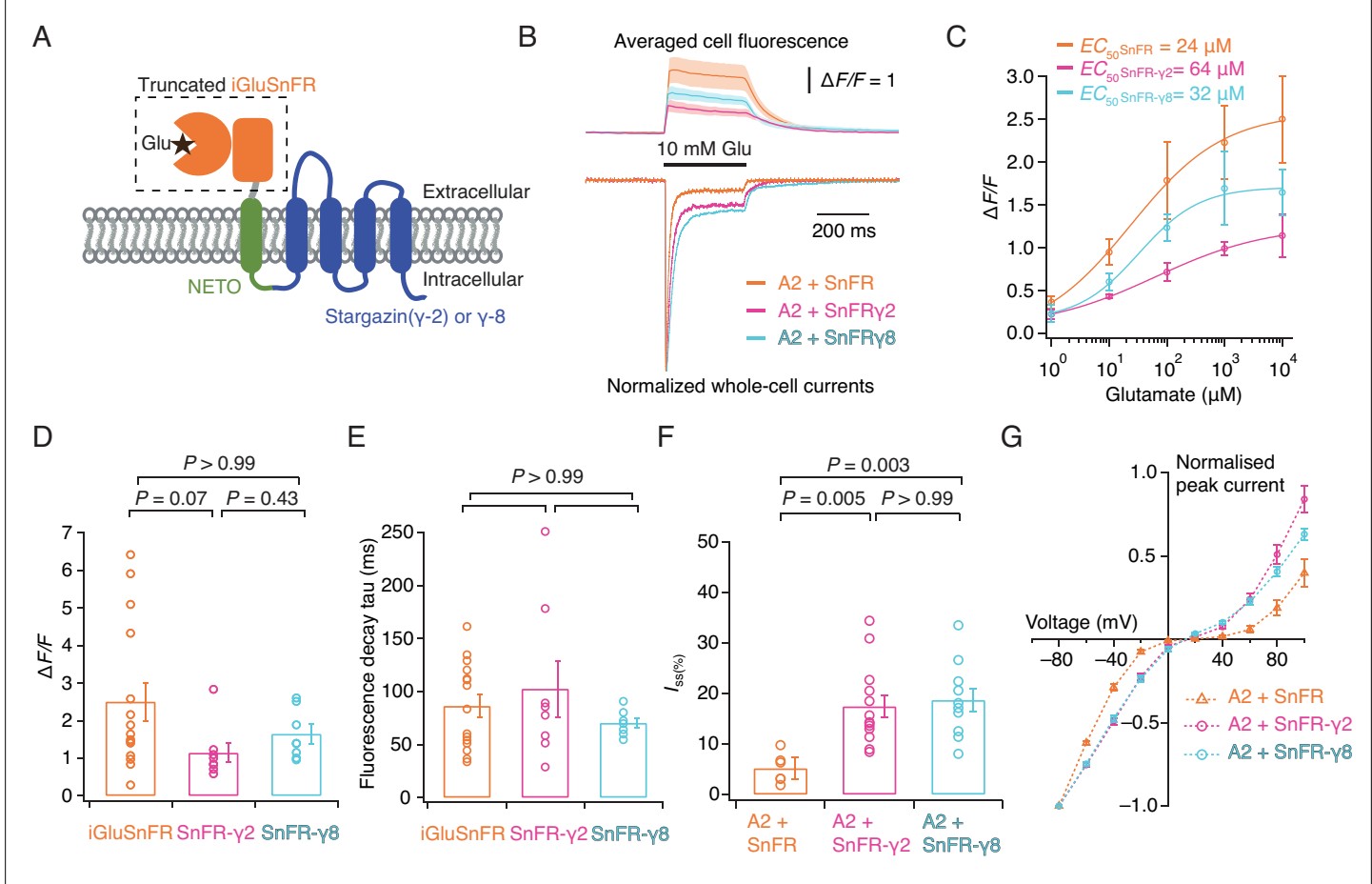

**Figure 1.** Patch-clamp fluorometry of SnFR-γ2 and SnFR-γ8 in HEK cells. (**A**) Schematic view of SnFR-γ2 and SnFR-γ8 chimeras, comprised of the truncated extracellular part of iGluSnFR (orange) to signal glutamate binding (black star), NETO as transmembrane linker (green) and Stargazin (γ-2) or γ-8 (blue) to act as a postsynaptic anchor. (**B**) Average fluorescent responses and normalised representative traces of currents for iGluSnFR (orange, $n = 15$), SnFR-γ2 (magenta, $n = 8$), and SnFR-γ8 (cyan, $n = 7$) during 10 mM glutamate application. GluA2 receptors were co-expressed with iGluSnFR, SnFR-γ2, or SnFR-γ8. The current recording from whole-cell patch-clamp fluorometry acts as a fiduciary for membrane expression, normal auxiliary protein function, and fast solution exchange. (**C**) Glutamate concentration–fluorescence response relationships for iGluSnFR ($n = 4$), SnFR-γ2 ($n = 4$), and SnFR-γ8 ($n = 5$) in HEK cells. (**D, E**) Statistics of peak fluorescent response and decay time constants (tau) for iGluSnFR ($n = 15$), SnFR-γ2 ($n = 8$), and SnFR-γ8 ($n = 7$), recorded as in panel (**B**). (**F, G**) Steady-state currents (normalised to peak response) and *I–V* relations for peak currents elicited by 10 mM glutamate for GluA2 and iGluSnFR cotransfection ($n = 6$), and for SnFR-γ2 ($n = 10$) and SnFR-γ8 ($n = 10$) complexes. Error bars represent standard deviation of the mean. Probabilities of no difference are from Dunn's non-parametric multiple comparisons test.

The online version of this article includes the following figure supplement(s) for figure 1:

**Figure supplement 1.** Kinetic properties and selectivity of iGluSnFR variants in HEK cells.

**Figure supplement 2.** Evoked currents and fluorescence signals of iGluSnFR, SnFR-Y230F variant, and synapto-pHluorin on autaptic neurons.

**Figure supplement 3.** Poor optical performance of truncated SnFR insertions to chimeric AMPA receptors.

Consistent with our results from immunocytochemistry (*Figure 3*), responsive SnFR-γ2 spots were enriched at Homer positive spots, presumably corresponding to synaptic connections (*Figure 4A*). Background-subtracted heat maps revealed a sharper signal for SnFR-γ2 compared to iGluSnFR (*Figure 4B*). Taking a line profile through the centre of the peak response showed that the apparent half-width of the responses during spontaneous neurotransmission was on average less than 500 nm (*Figure 4D, E*) or about three-fold narrower than for iGluSnFR in our cultures. When averaging across frames, the half-width at some sites was of the order of 300 nm, or in other words, probably limited by diffraction of the microscope (*Figure 4D*). For this spontaneous neurotransmission, mean amplitudes from both the peak and subsequent two frames from SnFR-γ2 were slightly increased compared to iGluSnFR (*Figure 4F, G*), possibly because SnFR-γ2 and SnFR-γ8

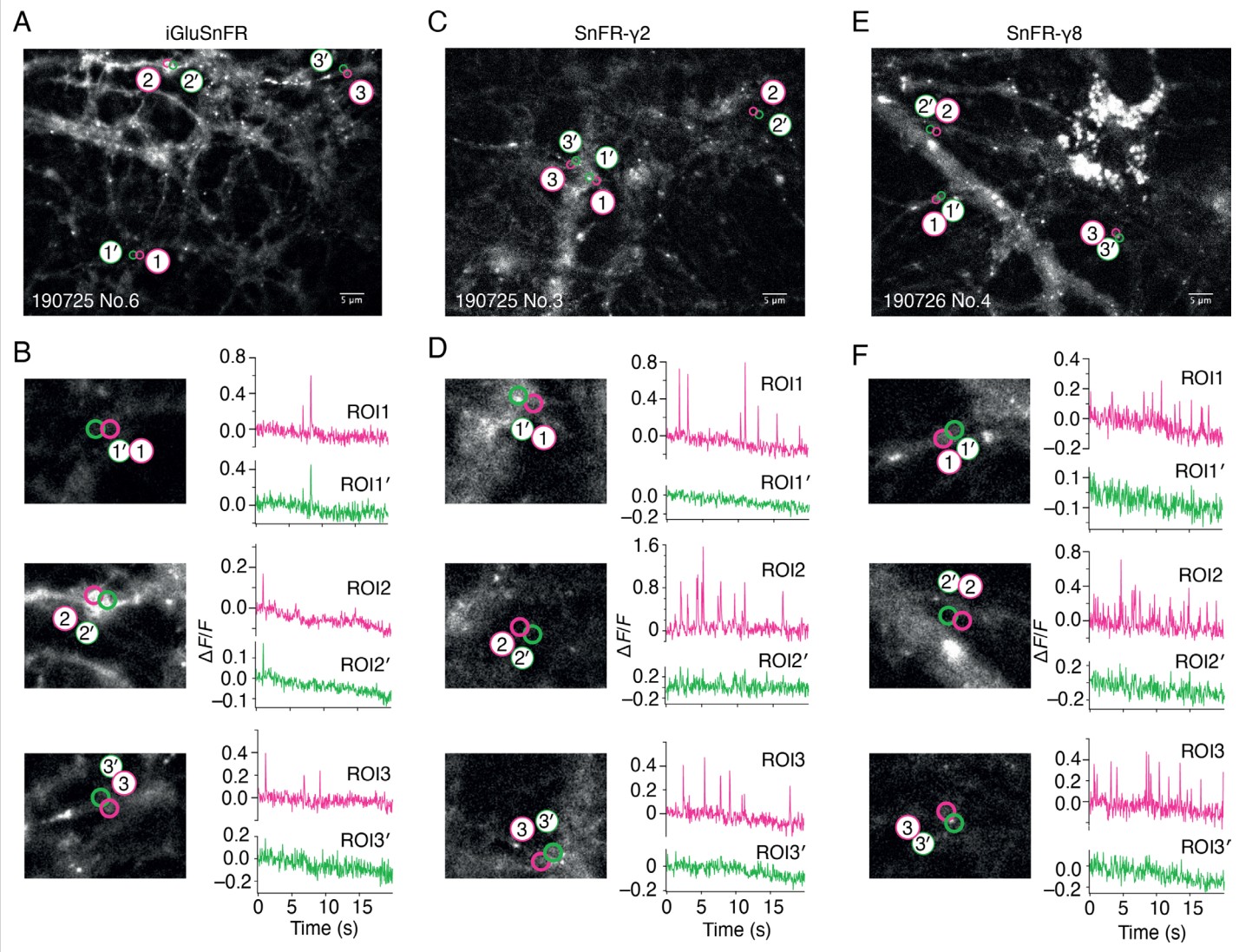

**Figure 2.** Spontaneous responses of iGluSnFR, SnFR-γ2, and SnFR-γ8 expressed in cultured rat hippocampal neurons. (**A**) Fluorescence micrograph of neurons expressing iGluSnFR with three typical regions of interest marked (ROIs, indicated as magenta circles) and compared with adjacent ROIs of the same size (shown as green circles, with prime). Scale bar, 5 μm. (**B**) Higher magnification views (2.2-fold zoom) around the individual ROIs and the corresponding colour-coded fluorescence time series. Vertical deflections correspond to spontaneous responses. Note similarity between magenta and green traces. All fluorescence time series were collected at 20 Hz. (**C, D**) As in panels A and B, but for neurons expressing SnFR-γ2. Note the lack of features in the green traces. (**E, F**) As in panels A and B, but for neurons expressing SnFR-γ8.

experienced a higher degree of saturation than their non-targeted version (see Discussion). The normalised peak fluorescence of the SnFR-γ2 response was also greater than that for iGluSnFR.

Infection of autapses with SnFR-γ2 and SnFR-γ8 gave punctate GFP signal in the majority of cases, but for some neurons, the signal remained diffuse. Although not obviously related, we also regularly measured a near absence of evoked neurotransmission (*Figure 5B*) in about half the autaptic neurons following infection before DIV 3. Robust evoked glutamate release (as reported by fluorescence changes from the sensors) was present, and we failed to detect any change in VGLUT puncta size or number on MAP2-positive neurites in immunostaining experiments (*Figure 5—figure supplement 1*), when comparing to non-infected neurons or neurons infected with iGluSnFR. These observations suggested a profound postsynaptic deficit due to overexpression of SnFR-γ2 and SnFR-γ8. We reasoned that, given that cultured neurons are still developing in this time window, we could infect the neurons later (after 6 days in vitro, DIV) and recover the evoked currents. Indeed, a majority of autaptic neurons retained an appreciable evoked response following late infection (neurons giving excitatory postsynaptic current

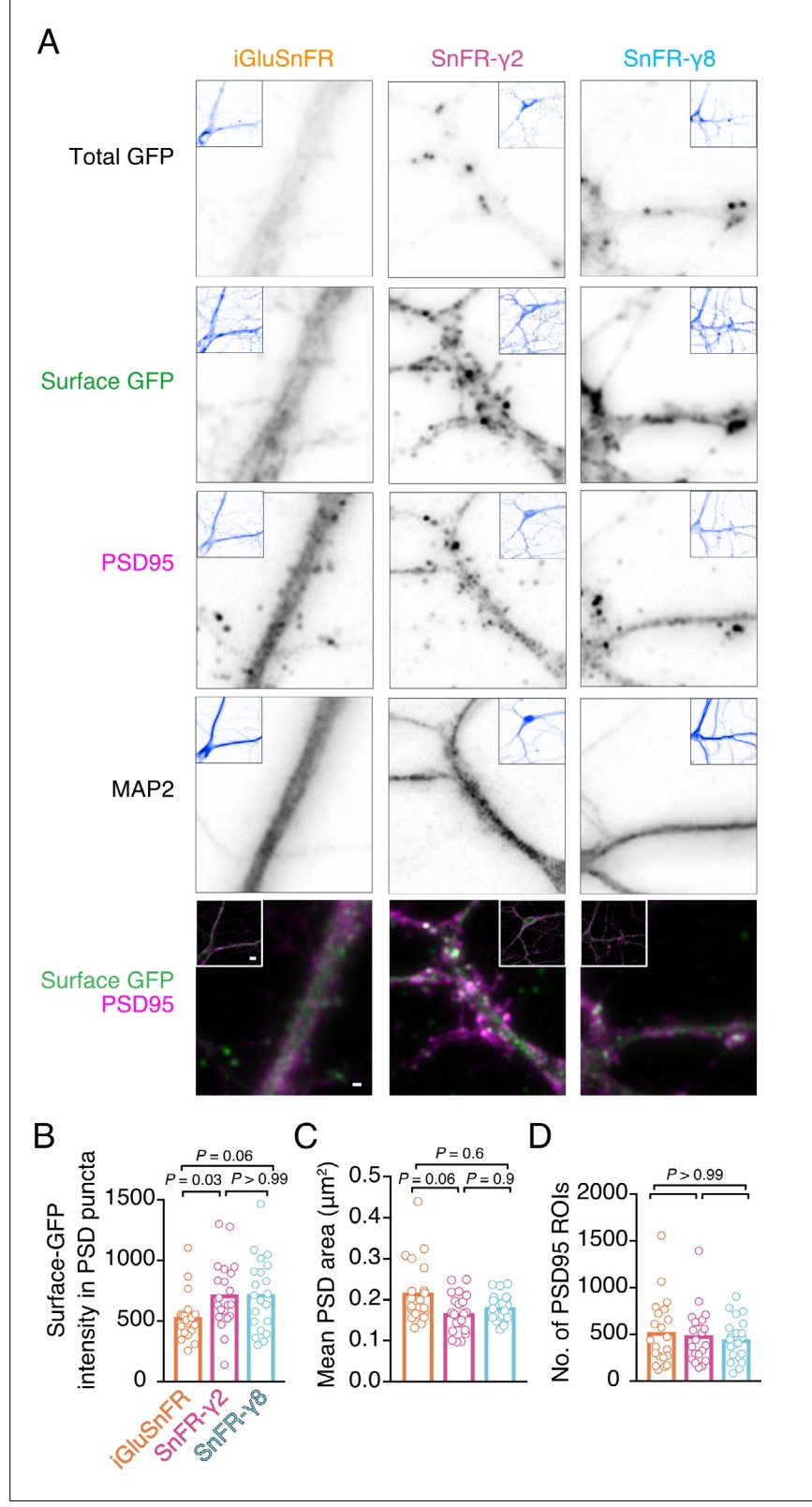

**Figure 3.** Localisation of iGluSnFR variants at synapses. (**A**) Representative images of hippocampal neurons infected with iGluSnFR, SnFR-γ2, and SnFR-γ8 and live immunostained with antibodies against GFP for surface labelling. After fixation, cells were stained for the postsynaptic marker protein PSD95 and neuronal marker MAP2. Top row shows the total GFP signal from the SnFR constructs in quiescent conditions. Bottom panels show

*Figure 3 continued on next page*

*Figure 3 continued*

merge of Surface GFP (green) and PSD95 (magenta) labelling for synaptic localisation. Scale bar, 1 μm. Inset images represent full field of view of each representative imaged neuron, with scale bar corresponding to 5 μm. Quantification of fluorescence intensity of surface GFP in PSD95 puncta, which was larger for SnFR-γ2 (**B**), mean area of PSD95 puncta on a per cell basis (**C**), and number of PSD95 puncta per field of view (**D**) in neurons infected with iGluSnFR, SnFR-γ2, and SnFR-γ8 (*n* = 21 neurons for each condition). Data shown are individual points and means. Three independent neuronal cultures were assessed, and probabilities of no difference were determined from Dunn's non-parametric multiple comparisons test.

[EPSC] >100 pA: 7 of 31, DIV 3 infection and 24 of 45, DIV 6; *Figure 5C*). The fluorescent signal and presynaptic markers (*Figure 5—figure supplement 2*) were unaffected by the late infection. Given that infection with iGluSnFR does not affect evoked currents much (*Figure 5C*), we reasoned that auxiliary proteins in overexpression might be causing the postsynaptic deficit. Indeed, early infection (before DIV 3) of either of two Stg constructs tagged with mScarlet (with or without the fused N-terminal NETO transmembrane section) were highly deleterious to evoked currents in autapses (*Figure 5F, G*). Miniature currents were also strongly reduced in amplitude and frequency, to the point that we failed to detect minis in a large proportion of cells (*Figure 5—figure supplement 2*), whereas paired-pulse ratio (20-ms interval) was unaffected. However, a late infection (after DIV 6) had much less impact, suggesting a previously unheralded developmental pathology at the postsynapse from early Stargazin overexpression.

In autaptic neurons infected with AAVs at DIV 6, the signal from iGluSnFR in response to evoked transmission at 5 Hz was again more diffuse than the response of SnFR-γ2 (*Figure 6*). We selected small ROIs and adjacent neighbour regions, again by hand. Whilst neighbour and principal fluorescence time series usually matched well for iGluSnFR (*Figure 6C*), for most ROIs in the SnFR-γ2 cells, the neighbour region was silent (*Figure 6F*). Correspondingly, the correlation coefficient between fluorescence time series for principal and neighbour ROIs was less for SnFR-γ2 (*Figure 6*).

Having established these prerequisites, we returned to our original motivation for improving the spatial and temporal response of iGluSnFR, which was to evaluate synaptic transmission at individual connections. We recorded movies of autaptic neurons infected with iGluSnFR (*Video 3*), SnFR-γ2 (*Video 4*), or SnFR-γ8. At the same time, we subjected the neurons to trains of 10 stimuli at 5 Hz in 0.5, 2, and 4 mM calcium. Escaping action potentials were generated from short depolarising pulses (see Methods), and we recorded the evoked postsynaptic currents under voltage clamp. From these movies, we examined the simultaneous evoked fluorescent responses from individual ROIs.

Rather than choosing ROIs by eye, we employed a custom-written imageJ macro to systematically scan movies (25-Hz frame rate) for ROIs that show similar statistics to the expected responses. Responses at 2 mM calcium were most reliable across the different sensors and we used these to

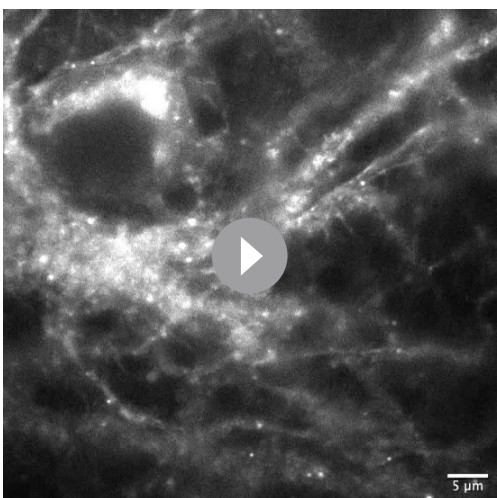

**Video 1.** Spontaneous responses of iGluSnFR expressed in bulk culture.

https://elifesciences.org/articles/84029/figures#video1

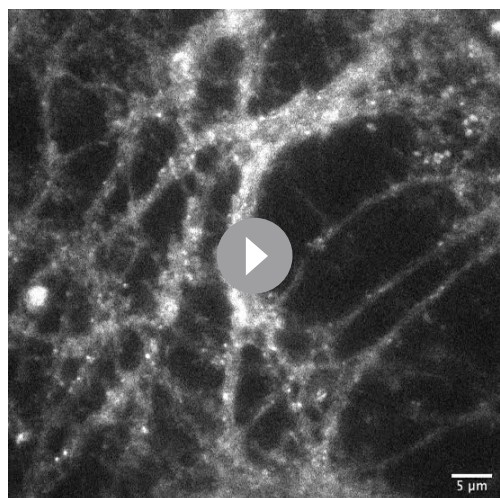

**Video 2.** Spontaneous responses of SnFR-γ2 expressed in bulk culture.

https://elifesciences.org/articles/84029/figures#video2

identify ROIs with >20 positive-going fluorescence transients per movie in which we made 50 stimulations. Composite ROIs were constructed from the contiguous regions identified. Strikingly, but perhaps unsurprisingly, the ROIs found in this way (*Figure 7*) were much smaller on average for SnFR-γ2 (mean areas: 2.5 vs. 9.6 µm$^2$) than for iGluSnFR. On average, neurons infected with iGluSnFR gave about twice as many ROIs as either SnFR-γ2 or SnFR-γ8 (*Figure 7C*; 207 ROIs for iGluSnFR, 107 for SnFR-γ2, and 97 for SnFR-γ8, each over 6 neurons). For all but one neuron, the number of ROIs from SnFR-γ8 was too few to pursue a further analysis.

Although iGluSnFR gave more ROIs, when we analysed the peak responses from each ROI, we saw that the responses of iGluSnFR in a subset of ROIs tended to run down (assessed as the magnitude of the last 10% of responses to the first 10%, *Figure 7—figure supplement 1*). This rundown represented as much as fourfold loss of the peak signal for some ROIs. Indeed at least half of automatically identified ROIs from the iGluSnFR movies showed rundown greater than 50% (*Figure 7—figure supplement 1*), whereas less than 7% of ROIs selected from SnFR-γ2 cells did. Classifying the ROIs in this way arguably lacks statistical rigour but indicates a large qualitative difference in the data obtained from the two sets of ROIs. There was no relation between the size of the compound ROI and the extent of rundown (*Figure 7—figure supplement 1*). Recordings were made on the same day with the same illumination conditions. Moreover, the average baseline fluorescence intensity ($F_0$) was stable during the recordings, and showed little change for either iGluSnFR or SnFR-γ2 in our recording conditions (*Figure 7—figure supplement 2*). This observation speaks against photobleaching as a source of the aberrant behaviour of iGluSnFR in some ROIs.

To understand if the rundown was a problem of the sensor, or perhaps simply an accurate report of the neurotransmission running down in these cells, we looked at how well the responses of the fluorescence reporters followed the summed postsynaptic electrical response to AP-induced release of glutamate, which we measured simultaneously. The EPSC peak amplitude typically reduced by about 40% over an entire run (see *Figure 7—figure supplement 2*), for both iGluSnFR and SnFR-γ2. Therefore, the iGluSnFR rundown exaggerated the current rundown and the SnFR-γ2 construct perhaps underestimated it, consistent with reporting different sources of glutamate (e.g., iGluSnFR preferentially reporting spillover). However, we note that changes in the peak response of only 30% are less easy to see in the noisy optical measurement (SNR ~3) than in the electrophysiology of evoked EPSCs (SNR ~1000).

Postsynaptic currents in iGluSnFR and SnFR-γ2 showed similar short-term plasticity (*Figure 8A, B*). With 2 mM calcium, the peak current amplitude exhibited mild depression during trains of five pulses. The initial amplitude was larger and showed a stronger reduction at 4 mM calcium, consistent with multivesicular release and vesicle depletion, whereas responses were small and did not change during the train for 0.5 mM calcium. We described the

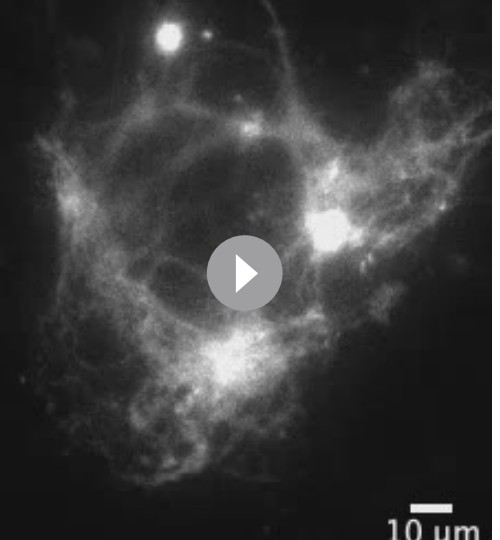

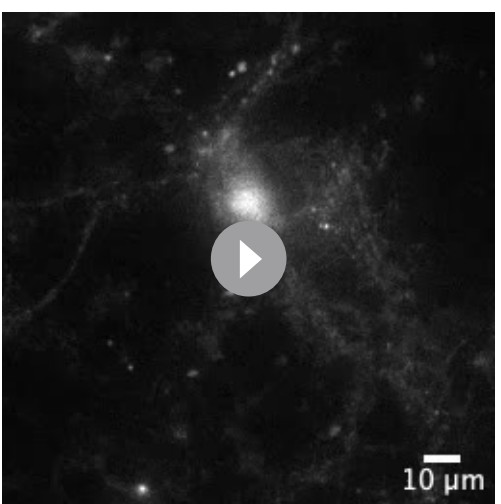

**Video 3.** Evoked responses of iGluSnFR expressed in autaptic neuron culture.

https://elifesciences.org/articles/84029/figures#video3

**Video 4.** Evoked responses of SnFR-γ2 expressed in autaptic neuron culture.

https://elifesciences.org/articles/84029/figures#video4

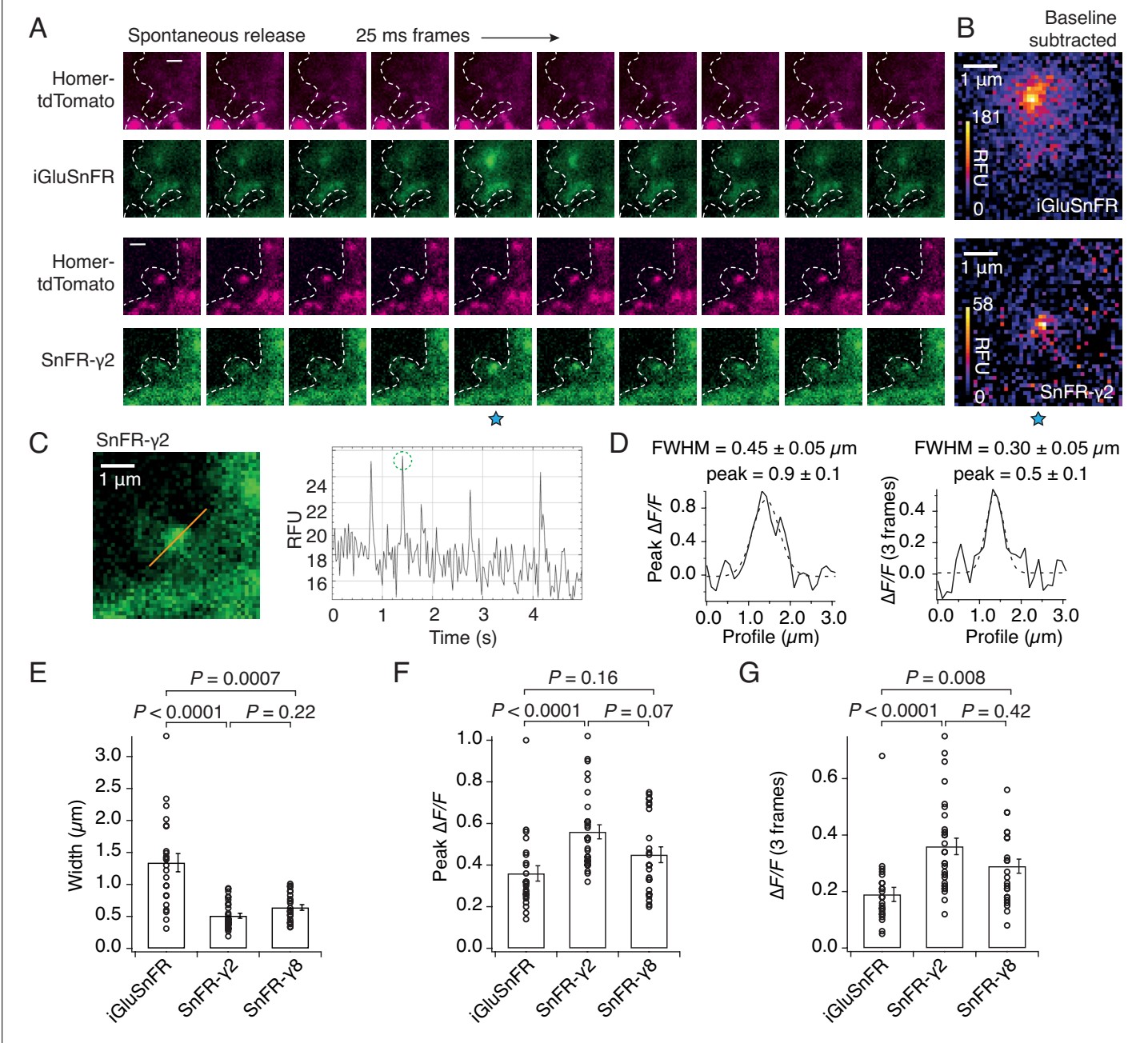

**Figure 4.** SnFR-γ2 and SnFR-γ8 give a more spatially precise signal than iGluSnFR. (**A**) Time-lapse with 40-Hz frame rate of representative spontaneous fluorescence responses for iGluSnFR and SnFR-γ2 (separate recordings from separate neurons). For SnFR-γ2, the signal coincides with Homer-tdTomato. The cell boundary (dashed line) was drawn by hand. Scale bars, 1 μm. (**B**) Background-subtracted heat maps of spontaneous fluorescence signals from the frame marked with the blue star. (**C**) Line profile (orange) and mean fluorescence time series from this line profile over 5 s. The second peak response was chosen for line profile and width analysis. (**D**) Fluorescence line profiles and Gaussian fits (dashed line) of individual peak response and average of the peak and the two consecutive frames. The widths of the fitted Gaussian profiles were 0.45 and 0.30 μm for individual peak and average three frames, respectively. (**E**) Fitted width profiles for SnFR-γ2 ($n$ = 30, 5 neurons) and SnFR-γ8 ($n$ = 25, 6 neurons) are substantially narrower than SnFR ($n$ = 24, 8 neurons). (**F**) SnFR-γ2 ($n$ = 30) showed larger relative fluorescence changes than SnFR ($n$ = 24) for the peak normalised fluorescence change. (**G**) Over the average three frames around the peak, both SnFR-γ2 and SnFR-γ8 showed larger responses. Data represent single fluorescent spots and mean ± standard deviation of the mean. Probabilities of no difference were determined from Dunn's non-parametric multiple comparisons test.

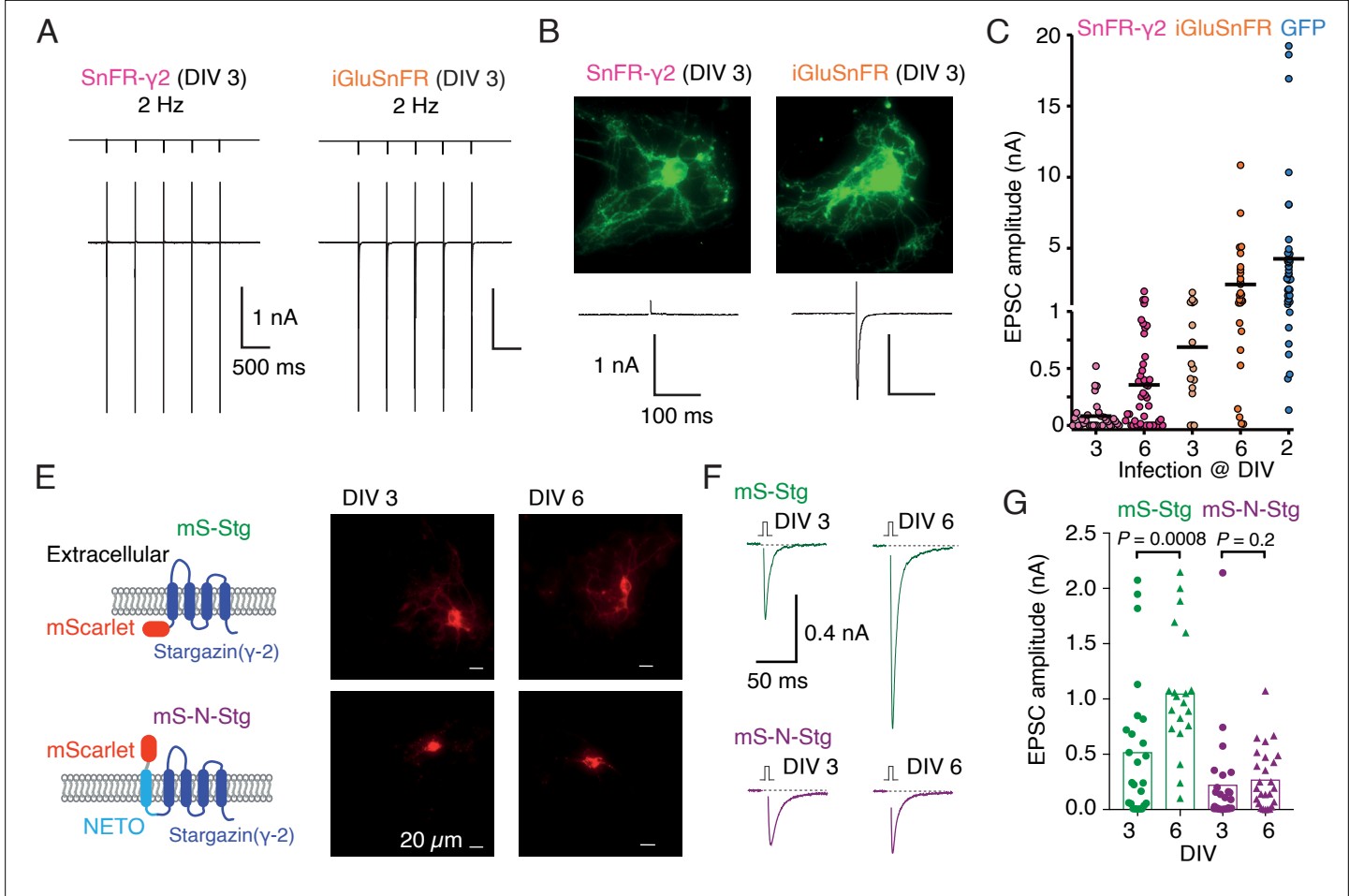

**Figure 5.** Early infection with SnFR-γ2 or Stargazin disrupts glutamatergic currents in autaptic hippocampal neurons. (**A**) Responses of autaptic neurons infected with SnFR-γ2 or iGluSnFR to a five pulse train at 2 Hz. (**B**) The maximum projections of the fluorescence *Eilers and Boelens, 2005* from each cell in panel (**A**), and the first response in the train with the action potential partially blanked. Both cells showed fluorescent responses to 2-Hz stimulation (not shown), but only the iGluSnFR-infected cell showed an excitatory postsynaptic current (EPSC). (**C**) Summary graph of the EPSC amplitudes in autaptic hippocampal neurons expressing SnFR-γ2 or iGluSnFR, infected on DIV 3 or before, and DIV 6 or after. EPSCs from cells infected with AAV-eGFP on DIV 2 are included for comparison (blue). (**E**) Cartoon of Stargazin constructs tagged on the N-terminal with mScarlet (mS-Stg, green, upper row) or extracellular mScarlet with NETO2 helix (mS-N-Stg, magenta, bottom row) and representative fluorescence images of autaptic hippocampal neurons infected at DIV 3 (left pair) or DIV 6 (right pair). (**F**) Representative EPSCs evoked by one action potential in autaptic hippocampal neurons expressing mS-Stg (green traces) or mS-N-Stg (magenta traces) infected at DIV 3 (left trace) or DIV 6 (right trace). (**G**) Summary graph of the EPSC amplitudes in autaptic hippocampal neurons expressing mS-Stg (green) or mS-N-Stg (magenta) at either DIV 3 (circles) or DIV 6 (triangles) (n = 25, 24, 20, and 26, respectively). Data shown are individual points, bars are means. Three independent neuronal cultures were examined, with recording and analysis performed blind. Probabilities of no difference are from the Mann–Whitney non-parametric test.

The online version of this article includes the following figure supplement(s) for figure 5:

**Figure supplement 1.** Neither SnFR-γ2 nor SnFR-γ8 influences synapse density or size in hippocampal neuron cultures.

**Figure supplement 2.** Electrophysiological parameters of autapses overexpressing Stargazin constructs.

degree of depression with the ratio of amplitudes in response to the first and last stimulation in the group ($Amplitude_5/Amplitude_1$). For fluorescence responses, we obtained peak responses from ROI time series following automatic (but supervised) background subtraction and discarded ROIs with a low peak signal-to-noise ratio (>3, see Methods).

In movies recorded from iGluSnFR expressing cells, there was no relation between the depression of the postsynaptic AMPA receptor response during each train and the average depression of the fluorescence signal over the ROIs (*Figure 8C*). In contrast, there was an excellent correlation between the SnFR-γ2 depression and the postsynaptic current depression at 2 and 4 mM calcium (*Figure 8C*) indicating that the synaptic release reported by SnFR-γ2 across tens of sites per neuron accurately

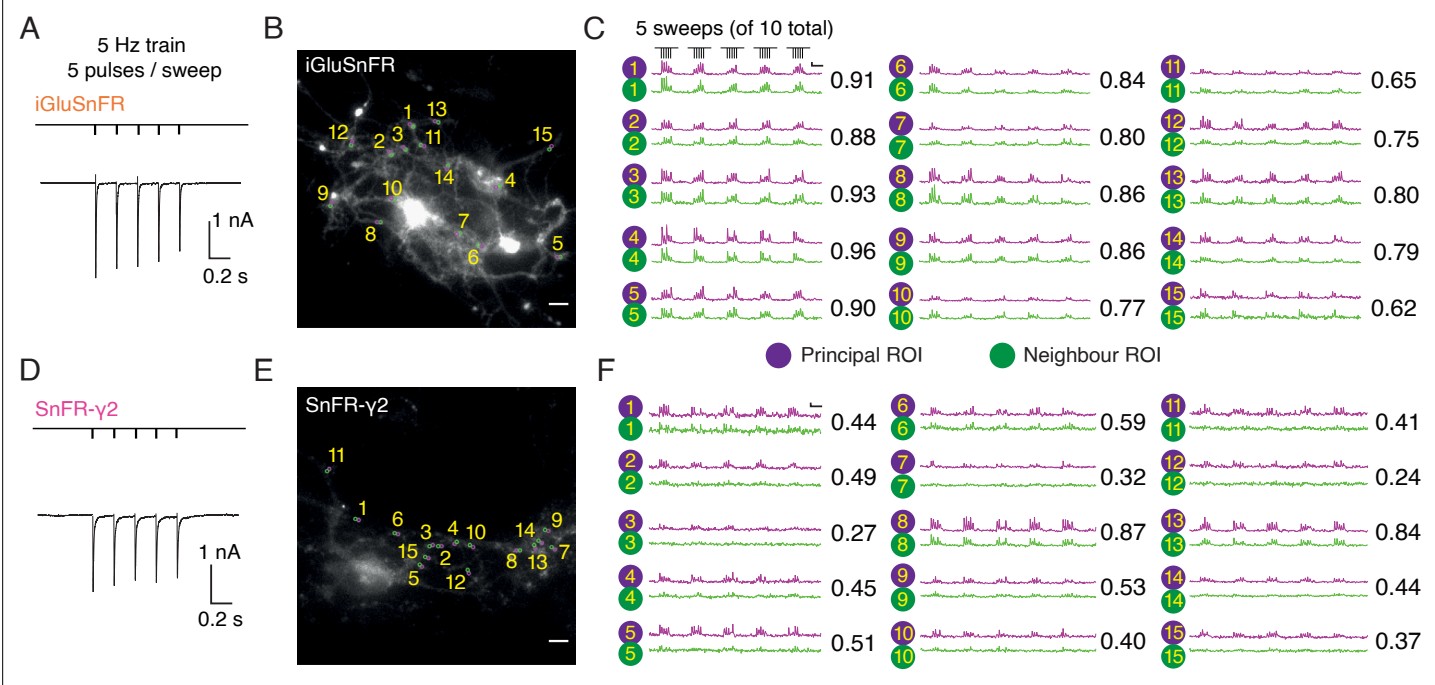

**Figure 6.** Evoked currents and fluorescent responses of iGluSnFR and SnFR-γ2 from autaptic neurons. (A) A train of five action potentials (APs) evoked five glutamatergic currents (excitatory postsynaptic currents, EPSCs) from an autaptic neuron infected with iGluSnFR (infected after DIV 6) in 2 mM calcium. The depolarisation and single AP are blanked to reveal the EPSC. This stimulus was repeated 10 times. (B) Fluorescence micrograph of the autapse recorded in (A) with 15 principal regions of interest (ROIs) (purple circles) against neighbouring ROIs of the same size (green circles). Scale bar, 5 µm. (C) Fluorescence time series from the principal and neighbour ROIs for five sets of stimuli (indicated above trace 1). Note similarity between purple and green traces in each case, indicating a broad signal. The correlation coefficient for each pair of optical recordings is indicated. For this set of recordings, the mean Pearson coefficient was 0.82 ± 0.02 for paired iGluSnFR ROIs and 0.48 ± 0.05 for paired SnFR-γ2 ROIs. Scale bar ΔF/F is 0.2 and timebase 1 s. (D–F) Equivalent data from an autapse expressing SnFR-γ2 (infected after DIV 6). Note that most green traces from neighbour ROIs do not respond (ROIs 8 and 13 are exceptions). Scale bars match those in panels (B) and (C).

represents the average synaptic response to glutamate release. At 0.5 mM calcium, one SnFR-γ2 cell gave an abnormal high potentiation (seven-fold). However, omitting this point did not reveal any correlation in the remaining data, so we kept this point in the dataset. The lack of correlation of signals from iGluSnFR is likely because of rundown and also possibly because indistinct localisation leads it to report stray glutamate as well as synaptic glutamate, particularly in conditions where repeated glutamate release occurs (5 Hz or above, see Discussion). The higher correlation in fluorescence between neighbouring sites that we observed for iGluSnFR (*Figure 6*) exemplifies this phenomenon.

Taking each train of five responses, we could plot the standard deviation of the five responses against the mean. In this way, we could estimate the coefficient of variation (CV) for a large number of synapses and responses (*Figure 8D*). Quantal analysis (*del Castillo and Katz, 1954*) suggests that the square of the CV is a simple function of the number of sites and the probability of release (see Methods). Even though we evoked EPSCs over presumably hundreds of connections, the selected ROIs allowed us to analyse the stoichiometry of release at putative single synapses optically. The number of ROIs is limited by the field of view and the focal depth of the objective lens. Taking the slopes from linear fits to paired SD/mean values, we performed a non-linear fit to the estimated squared CV values across the whole population of synaptic ROIs, allowing for a different release probability at each calcium condition, but common $N$ *(corresponding to the number of sites in an ROI)*. Most generally, $N$ here relates to the number of release sites per ROI (independent of the number of synapses in the ROI), and therefore $P_R$ is the neurotransmitter release probability at a single release site. Even though $N$ is averaged over a large number of ROIs (synapses), we fixed $N$ to be an integer to facilitate the search. For both conditions, $N$ could not be uniquely determined (three data points but four unknowns) and the data were fit equally well with a range of $N$ values. $N$ and $P_R$ are inversely correlated. However, for $N < 3$ the release probabilities were

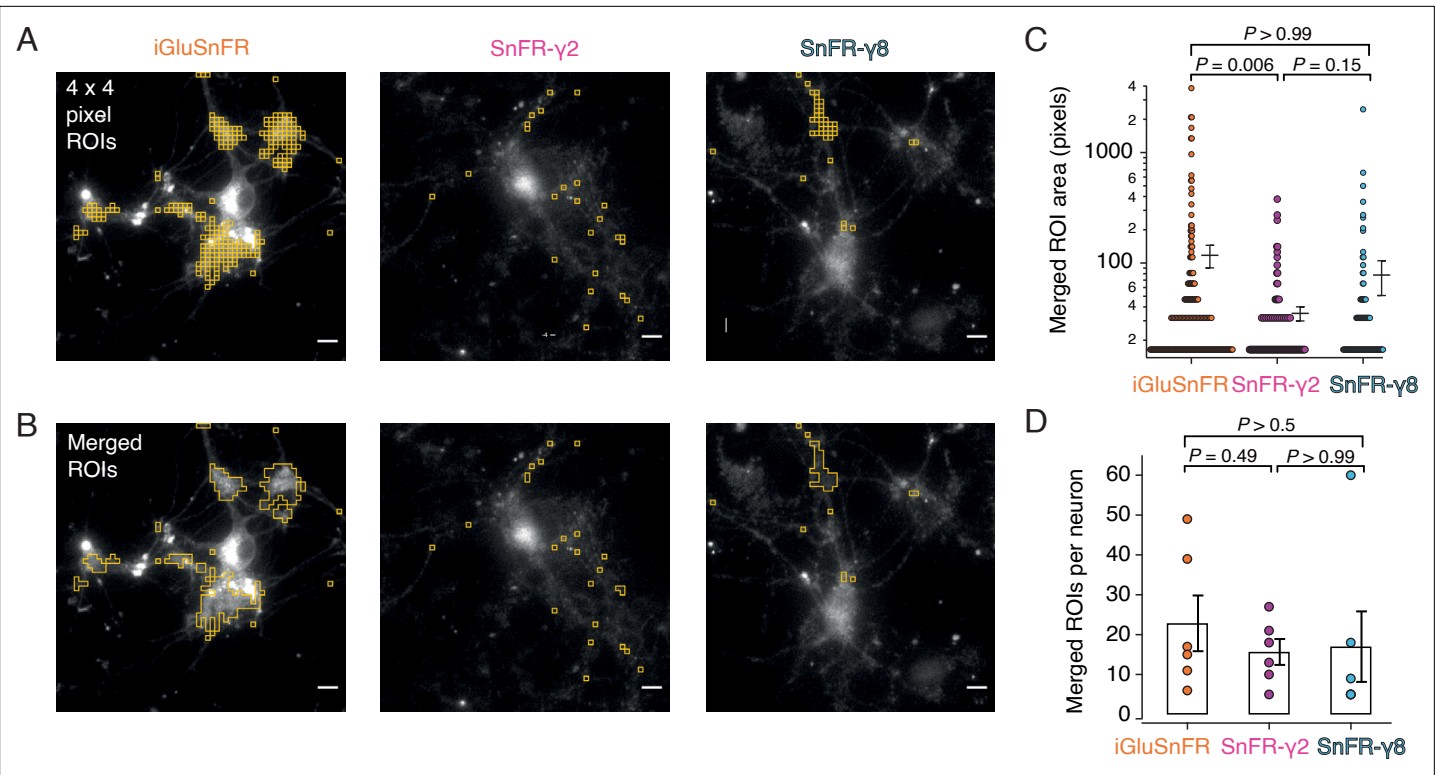

**Figure 7.** Targeted sensors give smaller and more reliable regions of interest (ROIs). (**A**) ROI size analysis for evoked fluorescent responses of SnFR, SnFR-γ2, and SnFR-γ8 on autaptic neurons. ROIs (orange outlines) automatically selected by a custom-written Fiji script (**Source code 1**) fulfilling the following conditions: 4 × 4 pixel ROI, intensity increase for each event ≥10%, SNR ≥3, and number of expected peaks in a movie >20. Scale bars, 5 μm. (**B**) As panel A but with contiguous ROIs merged. (**C**) Statistics of ROI sizes after merging for iGluSnFR (n = 207), SnFR-γ2 (n = 107) and SnFR-γ8 (n = 97) for six cells in each case. SnFR-γ2 shows smaller ROIs than iGluSnFR. (**D**) On average, the counts of ROIs per neuron were statistically indistinguishable between iGluSnFR, SnFR-γ2, and SnFR-γ8 (single field of view, ×60 objective). For panels C and D, probabilities of no difference were determined from Dunn's non-parametric multiple comparisons test.

The online version of this article includes the following figure supplement(s) for figure 7:

**Figure supplement 1.** Illustrative analysis of rundown of the fluorescence signal.

**Figure supplement 2.** Comparing rundown between electrophysiological responses and optical report.

unfeasibly high (**Figure 8E**), so we took a conservative approach, selecting the lowest N value that gave $P_R$ less than 50% for 0.5 mM Ca. Values of N above one could indicate either that each ROI contains multiple synapses, or, more likely given previous results on autapses, that the synapses show multivesicular release. As our results on individual ROIs below show, because practically no ROIs show N = 1 behaviour, the latter interpretation is also more likely here.

For CV analysis, independent of the choice of N value, the average release probability over all ROIs, estimated from SnFR-γ2, increased monotonically with $Ca^{2+}$ concentration, as expected from decades of electrophysiological analysis (**Figure 8G**, 6/6 cells). By contrast, the release probability estimated from iGluSnFR responses only increased monotonically for two out of six cells (**Figure 8G**), suggesting it is a poor reporter of synaptic activity in this context. These results indicate a clear quantitative benefit of targeted SnFR-γ2 over the non-targeted iGluSnFR. Previous analysis on connected cultured hippocampal neurons using electrophysiology gave similar values (N = 5 and $P_R$ = 0.4 from CV analysis **Bekkers and Stevens, 1990**) to the ones that we obtained from SnFR-γ2 at 2 mM $Ca^{2+}$. Although the congruence between connections between single neurons and the per-ROI parameters we found here is encouraging, we note that the CV method applied in this way cannot resolve N and therefore only gives an approximate rank order of synaptic release probabilities across the different conditions. We would therefore not recommend to use it.

In order to provide a more detailed analysis of synaptic release parameters, we next asked if SnFR-γ2 could be used to measure the quantal properties of individual synapses, and if it offered any

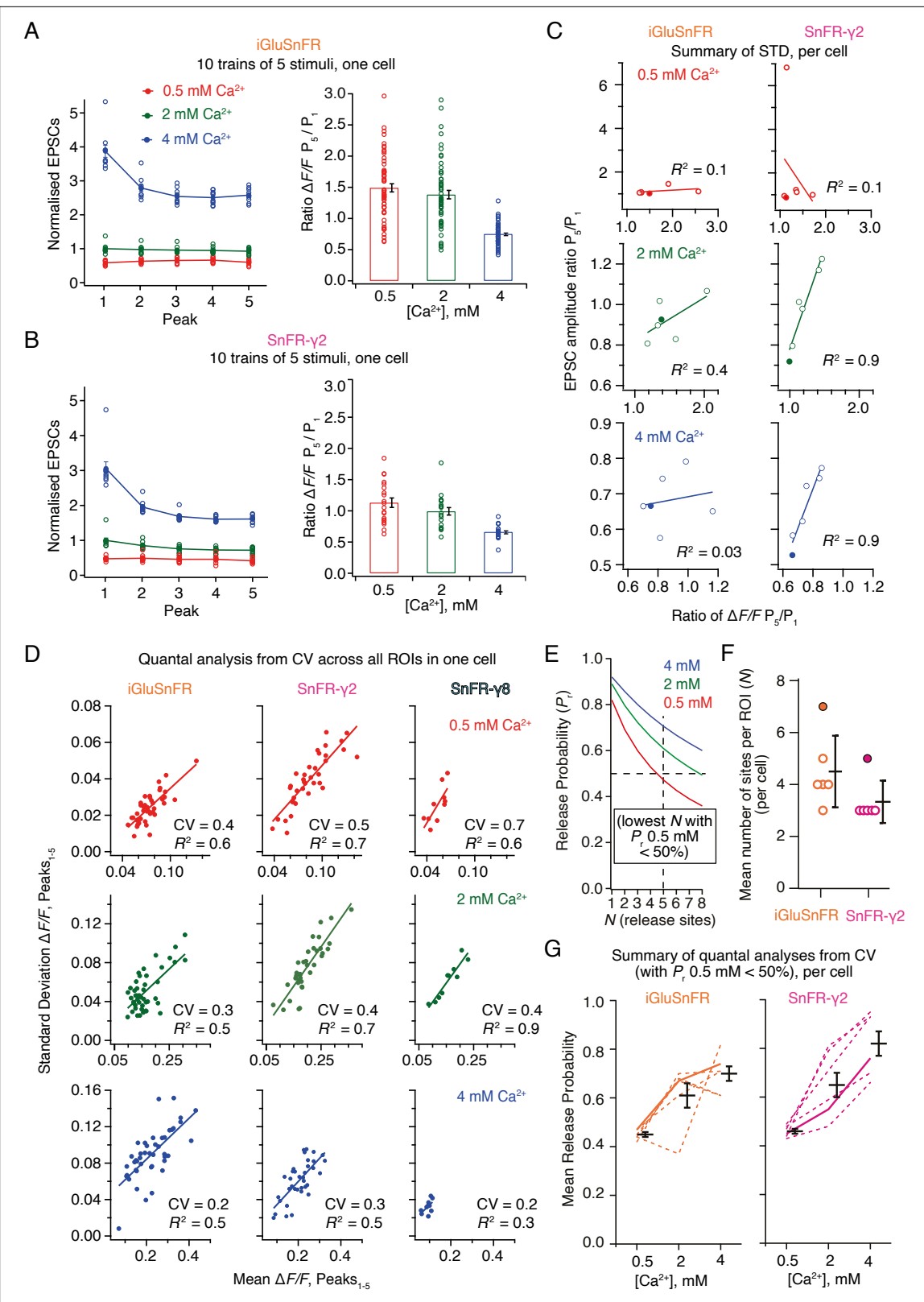

**Figure 8.** Correlations and variance of fluorescence responses. (**A**) Short-term plasticity of a train of five excitatory postsynaptic currents (EPSCs) evoked at 5 Hz for a single neuron (left, single responses to 10 trains and mean, normalised to the amplitude of the first response in 2 mM calcium). ΔF/F for multiple regions of interest in the same cell (right panel, bar shows mean and standard deviation of the mean). (**B**) As panel A but for a cell expressing SnFR-γ2. (**C**) Correlation between short-term depression (ratio between responses to fifth and first stimulus) from electrophysiology and fluorescence.

*Figure 8 continued on next page*

*Figure 8 continued*

The cells from panels (**A**) and (**B**) are indicated as filled circles. (**D**) To estimate population coefficient of variation (CV), the mean fluorescence response across 10 groups of 5 stimuli at 5 Hz is plotted against the standard deviation of these responses. Each point corresponds to a single region of interest (ROI). The data correspond to one cell at the three different calcium concentrations for each of the three different constructs. The CV (slope of the fitted line) was used in (**E–G**) to estimate population quantal parameters, except for SnFR-γ8 where the range was too small. (**E**) The goodness of fit was similar for a range of release site values (*N*, results of fit for SnFR-γ2 from panel D with *N* = 1–8 are shown) but only larger values gave release probability less than 50% for 0.5 mM Ca$^{2+}$. These generous criteria were used to select the most conservative *N* values. (**F**) The fewest effective release sites (per ROI) that gave release probability less than 50% for 0.5 mM Ca$^{2+}$ per cell. Solid symbols show the cells shown in panel D. Mean and standard deviation of the mean are indicated. (**G**) Dashed lines indicate release probabilities estimated over the population of responses for six cells (using *N* values as in panel F). Note that the increase in release probability was monotonic for SnFR-γ2 for every cell but only for 2/6 cells for iGluSnFR. The solid line indicates the cells in panel D. Mean and standard deviation of the mean are indicated.

advantages over the non-targeted iGluSnFR. To do this, we performed automated baseline subtraction from each ROI and detected peak responses according to a manually determined pattern from the mean of all responses. For both iGluSnFR and SnFR-γ2, we observed fluorescence time series with clear calcium dependence and quantal behaviour (*Figure 9—figure supplement 1*). From 50 stimulations at each concentration of calcium (150 events in total), at about half of the ROIs from iGluSnFR we could build histograms and fit globally (see *Figure 9—figure supplement 2* for workflow). For SnFR-γ2, 80% of the automatically-found ROIs were usable for quantal analysis, and these ROIs reported a range of release probabilities similar to those determined from the CV analysis (*Figure 9*). In the majority of ROIs, according to the Kolmogorov–Smirnov test for the goodness of fit, the binomial model gave a better fit than the Poisson model (*Bhumbra and Beato, 2013*; *Malagon et al., 2016*). Examples of imperfect fits are shown in *Figure 9—figure supplement 3*. For SnFR-γ2, the distributions of peak amplitudes gave clearer multiple peaks (*Figure 9C, D*) than iGluSnFR responses. However, these peaks were washed out above 3 or 4 components, even if up to 8–10 components were needed to cover the entire spread of amplitude data. The ranges of numbers of components and open probabilities determined from global fitting of individual synapses were slightly narrower for SnFR-γ2 than for iGluSnFR (*Figure 9F, G*).

In summary, the results from the SnFR-γ2 sensor were more convincingly quantal. Yet global fitting allowed similar, if less consistent, quantal parameters to be determined from iGluSnFR responses. Identification of single quanta at 0.5 mM Ca$^{2+}$ was challenging for both indicators, and this difficulty was more pronounced for SnFR-γ2, possibly because the smaller ROIs we obtained in this condition are inherently more noisy. This drawback meant that our quantal analysis (and particularly the estimation of the quantal size) relied on the equal spacing of the responses to multiple quanta at higher calcium concentrations. In turn, the best fits could only describe the histograms with limited accuracy (*Figure 9C, D*)—we expect that this could be improved by collecting more responses. The ROIs for iGluSnFR were on average much larger in size and number, and a larger fraction of them needed to be manually discarded because they showed strong rundown. A greater proportion of ROIs that were responsive for SnFR-γ2 could be described by quantal parameters, and very few were affected by rundown in the peak response that was seen in ~50% of the responsive ROIs in iGluSnFR-infected neurons.

## Discussion

The principal advantage of optical reporting of neurotransmission is direct access to individual synaptic responses. Taking optical responses from the field of view as group, SnFR-γ2 (but not iGluSnFR) accurately reported short-term depression, as measured simultaneously with electrophysiology, except in the noisy conditions of 0.5 mM calcium. On average, SnFR-γ2 also gave similar quantal parameters from global estimates of the CV to previous electrophysiological work (*N* = 5 and $P_R$ = 0.4 from CV analysis on pairs of cultured rat neurons; *Bekkers and Stevens, 1990*). These observations suggest that SnFR-γ2 is a fair reporter of presynaptic release, and that postsynaptic short-term plasticity (e.g., from desensitisation) is limited enough to be neglected at the frequencies we used (up to 5 Hz) (*Malagon et al., 2016*). However, both iGluSnFR and SnFR-γ2 failed to report gross variation in synaptic release properties across individual cells. Are synapses then truly variable (*Rosenmund et al., 1993*)? The deviation of individual mean/SD values from responsive ROIs (stemming from distinct synapses, either neighbouring groups or individual sites) from the fitted line representing

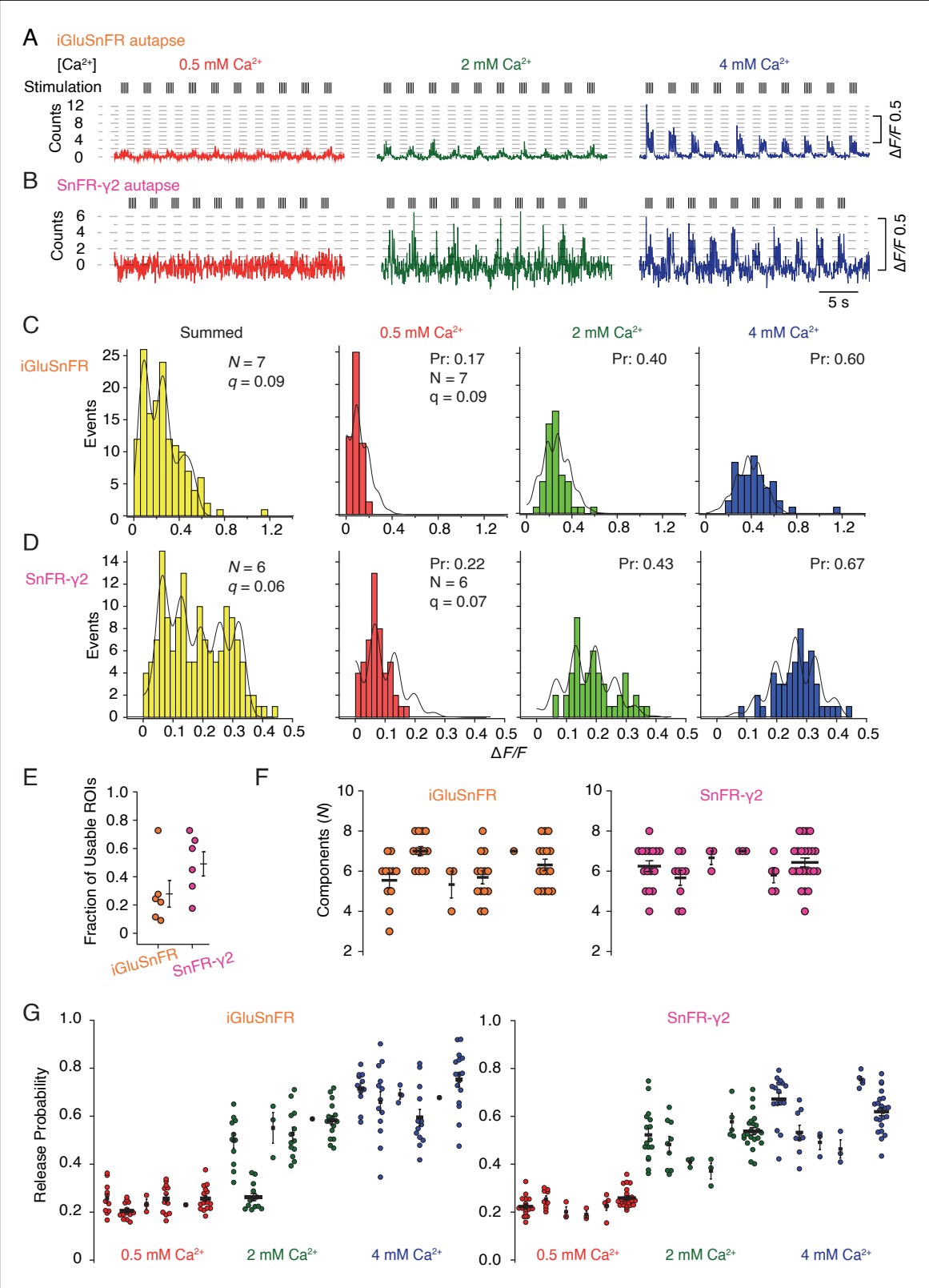

**Figure 9.** Quantal analysis in autaptic neurons. (**A**) Background subtracted fluorescence time series from one region of interest (ROI) from an iGluSnFR-expressing neuron at different calcium concentrations. Nominal quantal levels are indicated by dashed lines. The first response in 4 mM calcium is disproportionately large (and was usually ignored), and that steady-state responses accumulate during each train. (**B**) As in (**A**) but for a cell expressing SnFR-γ2, which showed smaller but more consistent signals. A detailed analysis of these two time series, and others recorded in parallel, is in **Figure 9—**

*Figure 9 continued on next page*

*Figure 9 continued*

***figure supplement 1.*** (**C**) Histograms of peak fluorescence responses from the iGluSnFR ROI in panel A. The summed histogram (yellow) represents 150 pooled responses, fitted with a mixed Gaussian model. Other histograms were separately fitted with the binomial model, where the quantal size (here $\Delta F/F$ = 9%) was globally optimised, with the scale. The release probabilities ($P_r$) were optimised for each calcium concentration. The number of release sites ($N$ = 7) was chosen to give the best fit (highest Kolmogorov–Smirnov probability). (**D**) As panel C but for the SnFR-γ2 ROI. (**E**) Only 25% of ROIs from the total were suitable for histogram fitting for iGluSnFR, whereas on average half of the SnFR-γ2 ROI were useable. (**F**) The distributions of $N$ chosen to give the best fit, from all the useable ROIs from each cell for iGluSnFR and SnFR-γ2 (six cells each). (**G**) The distributions of release probability for each ROI, grouped by cell and calcium condition.

The online version of this article includes the following figure supplement(s) for figure 9:

**Figure supplement 1.** Quantal nature of indicator responses.

**Figure supplement 2.** Workflow for peak response extraction and analysis of fluorescent report of glutamate release from synapses.

**Figure supplement 3.** Examples of imperfect fits to peak response histograms.

the global CV was limited, and the $R^2$ value was quite high for all conditions. However, the range of putative maximum release probabilities detected in individual cells was substantial (typically from 40% to 80%, at 4 mM calcium). One caveat is that our approach may be biased towards detecting synapses with higher release rates. Another important consideration is that the number of ROIs we typically observed (tens) is consistently fewer than expected from the number of synapses that would be expected to generate nA-scale evoked synaptic currents (hundreds). The reason for this disparity remains unclear. Correlation of the parameters of synaptic release determined from optical analysis to synaptic architectures, synapses altered by plasticity or experience and synapses whose properties are altered by pathological mutants, should yield information about cell-autonomous synaptic diversity (***Matz et al., 2010***).

Unlike at the neuromuscular junction (***del Castillo and Katz, 1954***), peaks in the histograms of evoked EPSC amplitudes are much less prominent at central synapses (***Bekkers, 1994***). Several mathematical treatments were proposed to deal with the smeared distribution that apparently comes from quantal variability at central synapses, including using Bayesian statistics and the gamma distribution (***Bhumbra and Beato, 2013***; ***Soares et al., 2019***). Optical report of glutamate concentration signal eliminates postsynaptic receptors as a source of variability. Miniature currents are variable in amplitude, introducing unavoidable variance when measuring evoked synaptic currents or when indirectly reporting activity from NMDA receptor calcium flux (***Oertner et al., 2002***). Simulations suggest that most variability comes from the variation in glutamate released by each vesicle (***Franks et al., 2003***), but we saw little evidence for this. Perhaps SnFR-γ2 is saturated by even poorly loaded vesicles, but fitting distributions with release sites ($N$) up to 8 or more, corresponding to up to 8 quanta in high $Ca^{2+}$, suggests this is not the case. AMPA-type glutamate receptors have a lower affinity for glutamate than SnFR-γ2, and so the invariant quantal size reported by SnFR-γ2 would in any case translate to less robust AMPA receptor activation. Synaptic AMPARs are probably often not saturated due to a steep dependence on geometric factors (***Savtchenko and Rusakov, 2014***), or because the number of AMPA receptors available is so variable. A similar "lack of dips" in histograms has been observed in distributions from iGluSnFR in postsynaptic sites (***Soares et al., 2019***; ***Heck et al., 2019***), requiring extensive mathematical modelling to extract quantal parameters. In clear contrast, confining iGluSnFR to the presynaptic terminal (in the case that anatomy allows this) gives quantal amplitude distributions (***Duerst et al., 2020***; ***Dürst et al., 2019***). In contrast with some previous work, for example using field stimulation (***Heck et al., 2019***), in our experiments in autapses, we stimulated glutamate release with a single action potential and obtained quantal histograms with separated peaks. In particular, we can confirm that changing the calcium concentration and examining the same terminal appears to be beneficial in determining quantal parameters (***Duerst et al., 2020***). On the other hand, in native tissue, the environment of synaptic terminals may reduce the ability of SnFR-γ2 to report quantal distributions, and future work should address this point.

Single AP stimulation enabled a clear view of the systematic fast loss of the iGluSnFR signal (***Figure 8***). Why does iGluSnFR run down and why does SnFR-γ2 instead have a stable response? We have no decisive mechanistic explanation. One possibility is that bleaching reduces $\Delta F/F$ because a large proportion of the background fluorescence signal comes from either intracellular or occluded sensor molecules. If bleaching (or another process that eliminates signal, like endocytosis) were more intense when glutamate is bound, and iGluSnFR were to in turn have more average occupancy by

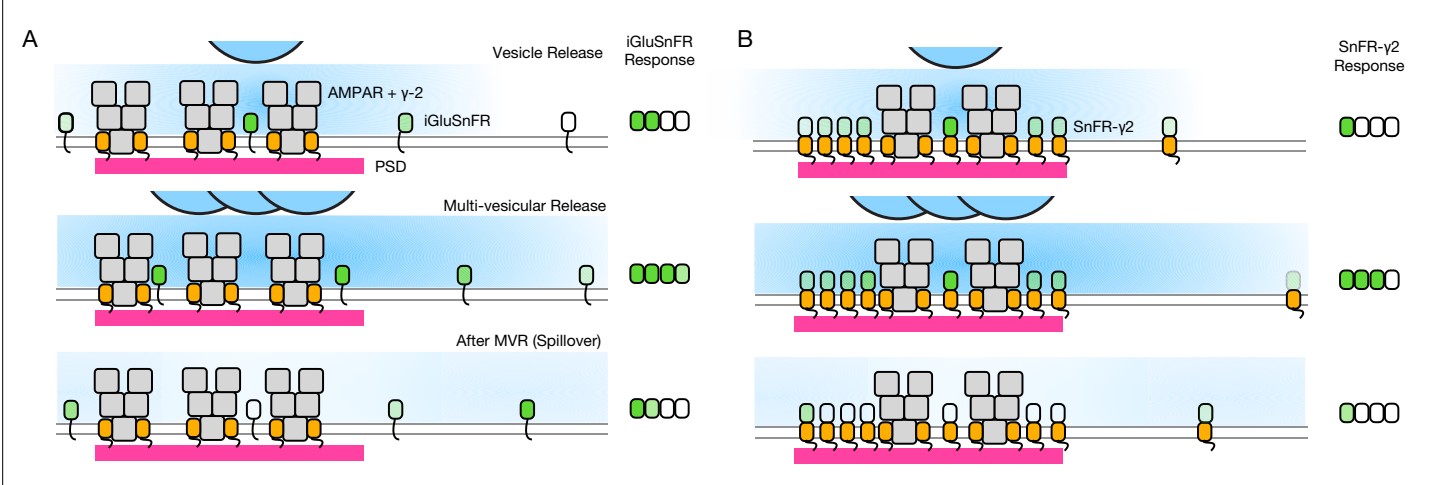

**Figure 10.** Factors affecting the responses of iGluSnFR and SnFR-γ2. (**A**) Three scenarios following glutamate release from vesicles (blue). iGluSnFR is not localised to the postsynaptic density (PSD, magenta) opposite release sites, where AMPA receptors are docked through their auxiliary proteins like γ-2 (orange). Therefore iGluSnFR integrates a wide glutamate signal (green response) because of its high affinity for glutamate compared to AMPA receptors. This summation gives a larger amplitude signal, and probably incorporates a robust response to extrasynaptic spillover after MVR. Note, in a fluorescence micrograph, the point spread function is larger than the scene depicted here. Also, the camera integration time is longer than the interval between MVR and spillover. These factors do not provide an obvious explanation for the rundown of SnFR responses. (**B**) Assuming that SnFR-γ2 is concentrated at synapses, relative to iGluSnFR, the principal contribution following release of either one vesicle or multiple vesicles (MVR), is from synaptic sites. This limits the signal in amplitude, in space, and in time. SnFR-γ2 probably competes with postsynaptic AMPA receptors for sites in the PSD. The ratio of extrasynaptic to synaptic membrane area is large, we do not intend to indicate a ratio between abundances at these different sites. Overall, SnFR-γ2 is more abundant outside synapses than within them.

glutamate than the confined SnFR-γ2, this could specifically reduce Δ*F*. In such a scheme, SnFR-γ2 would be protected from glutamate exposure by seeing only fast glutamate transients that saturate the sensor and immediately disperse, as opposed to a larger iGluSnFR capacity that is saturated for longer periods by spillover (*Figure 10*). The diffusion within synapses of the same single TM protein with and without a PSD-binding ligand (from the Stg C-terminus) provides an excellent comparison for our purposes (*Li and Blanpied, 2016*). These studies show that even though the diffusion of any protein is impeded within the PSD, the membrane protein without any PSD ligand is not excluded. However, our results are consistent with the idea that the PDGFR motif of the original iGluSnFR is excluded from synapses, as corroborated by expansion microscopy (*Aggarwal et al., 2022*).

Assuming equal expression, and the concentration of membrane proteins with a PDZ ligand inside PSDs (*Li and Blanpied, 2016*), the density of SnFR-γ2 must be less outside the synapse (also see *Figure 6*). On average, a greater capacity outside the synapse would mean that, even without being excluded from synapses, iGluSnFR is biased against measuring synaptic glutamate. The basal response of iGluSnFR might be greater in neurons (even though in HEK cells the response was similar) because it is always moving in relation to release sites, and can respond to distant glutamate better by covering greater areas (even within an ROI).

Neither iGluSnFR nor SnFR-γ2 can identify single release events in our conditions, but for different reasons. The iGluSnFR signal often runs down intensely, meaning that the quantal estimate Δ*F* changes during a recording and cannot be considered constant across different conditions. SnFR-γ2 cannot reliably report isolated single release events because the amplitude of the signal is too small. However, spontaneous neurotransmission in cultures (in the absence of TTX) could be readily resolved by SnFR-γ2, at spot-like ROIs. These optical responses may correspond to single or multiple vesicles released by action potentials. For short-term plasticity, whereby synapses are reactivated over 50-ms intervals or similar, the improved spatial profiles of SnFR-γ2 are expected to be beneficial (*Helassa et al., 2018*). In our experiments, we did not focus on ROIs with high-frame rate, but this is an obvious future application.

iGluSnFR has been used in a variety of organisms (*James et al., 2019*; *Marvin et al., 2013*; *Borghuis et al., 2013*), and here we expressed SnFR-γ2 only in cultured rodent neurons. Using mouse auxiliary proteins to anchor the glutamate reporter opposite to release sites may not be universal, and may

influence synaptic transmission in other systems in undesirable ways. Stargazin overexpression reduced miniature current amplitude and frequency but did not affect paired-pulse response (*Figure 5—figure supplement 2*) and glutamate release was intact (from SnFR-γ2 signals). This strongly suggests a post-synaptic deficit in cultured hippocampal cells but in other neuronal types, effects might be different. When we made inadvertent recordings of GABAergic cells, which we subsequently discarded, we saw no response of the iGluSnFR sensors, but nor was there any obvious effect on evoked inhibitory current magnitudes. It is unclear how well this approach can translate across species, even though PSD binding should be conserved. Notably, sensor performance is always worse in the in vivo context, and our sensor, along with a similar recently reported approach (*Aggarwal et al., 2022*), is probably better suited to quantitative studies of synaptic transmission in vitro, rather than in vivo work, even though its performance should not be worse than the original iGluSnFR construct.

Why does overexpressing various forms of Stargazin in the first few days of culture create a specific postsynaptic deficit? Stargazin has a very long first intron (*Letts et al., 1998*). This long first intron of Stg is the target of three mutations (including the original stargazer mouse) which reduce or abolish γ-2 mRNA (*Letts et al., 2003*). Because long introns extend transcription times, they are thought to delay expression. The most curious aspect of our observations was that release was apparently normal, and presynaptic VGLUT1 markers were not affected. It is not routine to perform electrophysiological characterisation of synapses following molecular interventions before imaging, however, the synapses in neurons infected with Stargazin or chimeras on or before DIV 3 were non-functional, despite their otherwise normal appearance. It is likely that Stargazin is not the ideal host for the iGluSnFR sensor domain. Auxiliary proteins that are not complexed to AMPA receptors probably fill slots in the postsynaptic density, reducing postsynaptic currents. The performance of the SnFR-γ8 construct was similar to SnFR-γ2, but neuronal labelling was often less broad, giving fewer ROIs that responded. It is likely that tethered sensors preferentially select stronger synapses. The additional transmembrane helix from NETO2 appears to reduce expression (although this may indeed be desirable) and is quantitatively more damaging to synaptic transmission than Stargazin in a more native form (*Figure 5*).

Rather than obtaining an optimised sensor, we have instead demonstrated further proof of principle that postsynaptic tethering improves reporter characteristics (*Soares et al., 2019*), even if it reduces overall signal. Our targeted sensor did not have distinctly better fluorescence performance in terms of signal to noise compared to iGluSnFR. A recent report used the principle of tethering to localise the signal (*Aggarwal et al., 2022*). One disadvantage of targeting may be that non-synaptic signals of interest could be lost. More critically, the synaptic localisation does not guarantee to measure exclusively synaptic glutamate; waves of elevated glutamate from other sources including astroglia, substantial spillover or even changes in the optical properties of tissue could be mistaken for synaptic glutamate in this context, and this is a limitation of our approach. In organised brain tissue, two-photon excitation will not avoid this difficulty. Spectral variants with distinct (or no) targeting could be combined to get a fuller picture of both synaptic and spillover glutamate (see *Hao and Plested, 2022* for discussion). Various approaches are known to augment the signal from iGluSnFR including using the superfolder variant of GFP (*Marvin et al., 2018*), mutagenesis (*Figure 1—figure supplement 1*), or a different promoter (we used human Synapsin). Indeed, the superfolder variant of iGluSnFR could be used very successfully for quantal analysis in cell culture, when analysis was confined to axonal projections (i.e., presynaptic sites, *Mendonça et al., 2022*). More work is needed to understand why targeting reduced the size of ROIs at postsynaptic sites in our hands, whereas well-confined signals are seen in axons. Finally, other tethers, either minimal ones (*Li and Blanpied, 2016*) or those based on other synaptic proteins, may also give better, more defined, or even orthogonal signals.

# Materials and methods
## Materials
All chemicals were purchased from Sigma-Aldrich unless otherwise stated. MEM Eagle was from PAN-Biotech. Fetal bovine serum (FBS), trypsin, and penicillin/streptomycin, Neurobasal-A, B27, Glutamax, and gentamicin were from Thermo Fisher. All DNA restriction enzymes and T4 ligase were from Thermo Fisher. dNTP set was from Qiagen. Plasmid purification kits were from ROBOKLON.

## Cell lines

HEK 293 cells (ACC 305) were purchased from Leibniz Institute DSMZ (German Collection of Microorganisms and Cell Cultures GmbH, Braunschweig). According to the DSMZ, STR analysis according to the global standard ANSI/ATCC ASN-0002.1-2021 (2021) resulted in an authentic STR profile of the reference STR database. In-house testing for mycoplasma yielded a negative result.

## cDNA constructs and molecular biology

pCMV(MinDis).iGluSnFR was a gift from Loren Looger (Addgene plasmid # 41732; http://n2t.net/addgene:41732; RRID:Addgene_41732) (*Marvin et al., 2013*). pmScarlet_C1 was a gift from Dorus Gadella (Addgene plasmid # 85042; http://n2t.net/addgene:85042; RRID:Addgene_85042) (*Bindels et al., 2017*). pAAV-CW3SL-EGFP was a gift from Bong-Kiun Kaang (Addgene plasmid # 61463; http://n2t.net/addgene:61463; RRID:Addgene_61463). mEos3.2-Homer1-N-18 was a gift from Michael Davidson (Addgene plasmid # 57461; http://n2t.net/addgene:57461; RRID:Addgene_57461). CMV::-SypHy A4 was a gift from Leon Lagnado (Addgene plasmid # 24478; http://n2t.net/addgene:24478; RRID:Addgene_24478). GluA2 in the pRK5 vector was the kind gift of Peter Seeburg and Mark Mayer. Stargazin, NETO2 and gamma-8 were kind gifts from Susumu Tomita and Roger Nicoll.

Mutations (including Y230F, *Figure 1—figure supplement 1*) were introduced by overlap PCR to generate iGluSnFR variants. For generation of SnFR-γ2 and SnFR-γ8, the Myc tag and PDGFR transmembrane domain were removed from iGluSnFR and replaced by a 12 amino acid linker (GGRARADVYKRQ) followed by NETO2-γ2 or NETO2-γ8 chimeras. NETO2-γ2/NETO2-γ8 chimeras were composed of a 79 amino acid stretch of rat NETO2 including the TM segment (starting with residue E333 of Uniprot C6K2K4) and rat Stargazin/gamma-8 with a 3 amino acid (AGS) linker between the two. The entire cassette was subcloned into the pcDNA3.1+vector. For overlapping PCR of recombinant SnFR-γ2/SnFR-γ8, we used forward primer 5′ CATTGACGCAAA TGGGCGGTAG 3′ and reverse primer 5′ CCGGGCGCGCCCACCTTTCAGTGCCTTGTCATTCGG 3′ for the iGluSnFR fragment, and forward primer 5′ CCGAATGACAAGGCACTGAAAGGTGGGC GCGCCCGG 3′ and reverse primer 5′ CAACAGATGGCTGGCAACTA 3′ for NETO2_γ-2/NETO2_γ-8 fragments. To make the mScarlet-NETO-Stg construct we replaced the iGluSnFR domain (Glt1 and cpGFP) in SnFR-γ2 with mScarlet, retaining the IgK signal to ensure extracellular topology of the N-terminus. For the mScarlet-Stg construct we added mScarlet at the N-terminus of Stargazin. For overlapping PCR of recombinant mScarlet-stg or mScarlet-NETO2-stg, we used forward primer 5′ CATTGACGCAAATGGGCGGTAG 3′ and reverse primer 5′ CTTATACACATCTGCCCGGG CGCGCCCACCCTTGTACAGCTCGTC 3′ for the mScarlet fragment, and for Stargazin or NETO2-stargazin fragments we used forward primer 5′ ACGAGCTGTACAAGGGTGGGCGCGCCCGGGCA GATGTGTATAAGAGACA 3′ and reverse primer 5′ CAACAGATGGCTGGCAACTA 3′. For electro-physiological studies on HEK cells we used the pRK5 expression vector encoding the flip splice variant of the rat GluA2 subunit containing a Q at the Q/R-filter, and removed the IRES-eGFP that followed the GluA2 cDNA. The following plasmids are deposited at Addgene: SnFR-gamma2 (Addgene ID: 165495), SnFR-gamma8 (165496), pAAV-syn-SnFR-gamma2-minWPRE (165497), and pAAV-syn-SnFR-gamma8-minWPRE (165498).

## Lentiviral and adeno-associated constructs and virus production

Expression vectors for proteins of interest were delivered to primary neurons by either lentivirus or AAVs. The lentivirus construct for f(syn)-Homer1-tdtomato (BL-1034 in the Charité VCF catalog) was a subclone of mEos3.2-Homer1-N-18. The lentivirus construct for rat Synaptophysin-pHluorin (*Granseth et al., 2006*) was a subclone of Addgene #24478 (BL-0047 in the Charité VCF catalog). For AAV production, each expression cassette (iGluSnFR, iGluSnFR Y230F, SnFR-NETO2-γ2, SnFR-NETO2-γ8, IgK-mScarlet-NETO2-γ2, or mScarlet-γ2) was subcloned into a pAAV-backbone with a WPRE-enhanced sequence (*Choi et al., 2014*) and a synapsin promoter. Preparation of lentiviral particles and AAV were performed by the Charité Viral Core Facility (Charité-Universitätsmedizin, Berlin). Lentiviral particles were prepared as previously described (*Lois et al., 2002*). Briefly, HEK293T cells were co-transfected with the lentivirus shuttle vector (10 µg) and two helper plasmids, pCMVdR8.9 and pVSV.G (5 µg each) using polyethylenimine (PEI). After 72 hr, virus-containing supernatant was collected, filtered, aliquoted, and flash-frozen with liquid nitrogen. Virus aliquots were stored at

−80°C. For infection, about $5 \times 10^5$ to $1 \times 10^6$ infectious viral units were pipetted onto WT hippocampal neurons per 35-mm-diameter well. AAV was prepared as described previously (*Rost et al., 2015*).

## Mammalian cell culture and transfection

HEK293 cells for electrophysiological experiments were cultured in Minimum Essential Medium (MEM) supplemented with 10% (vol/vol) serum, 5% U/ml penicillin, and 100 µg/ml streptomycin at 37°C in a humidified 5% $CO_2$ environment. HEK293 cells were transiently transfected using PEI in a 1:3 ratio (vol/vol; DNA/PEI) with OptiMEM 1 day after cells were seeded. The ratios for co-transfection were 1:1 for GluA2 and iGluSnFR, 1:2 for GluA2 and SnFR-γ2, and 1:5 for GluA2 and SnFR-γ8, up to 3 µg total DNA per 35 mm dish. After 5 hr of incubation, the transfection medium was replaced by fresh MEM supplemented with 40 µM NBQX with the aim of reducing TARP (γ-2/γ-8)-induced cytotoxicity.

## Neuronal culture and transduction

For primary bulk culture of hippocampal neurons, hippocampi were dissected from 1- to 3-day-old rat brains, cut into small pieces in the dissection medium (Hank's modified solution with 4.2 mM $NaHCO_3$, 12 mM 4-(2-Hydroxyethyl)piperazine-1-ethane-sulfonic acid (HEPES), 33 mM D-glucose, 200 µm kynurenic acid, 25 µm 2-Aminophosphonovaleric acid (APV), 5 µg/ml gentamicin, 0.3 mg/ml bovine serum albumin [BSA], 12 mM $MgSO_3$, pH 7.3) and digested in the digestion medium (137 mM NaCl, 5 mM KCl, 7 mM $Na_2HPO_4$, 25 mM HEPES, 4.2 mM $NaHCO_3$, 200 µm kynurenic acid, 25 µm APV, 0.25% trypsin, and 0.075% DNAse, pH 7.4) for 5 min at 37°C, and then blocked with the dissection medium containing 0.1% trypsin inhibitor for 10 min at 4°C and dissociated with the dissociated medium (50 ml Neurobasal-A, 0.5 ml Glutamax 100×, 2 ml B27, 0.625 ml FBS) to acquire isolated neurons. Cultured neurons were incubated at 37°C, 5% CO2 in a culture medium composed of Neurobasal-A containing 2% B27, Glutamax (10 mM) and Gentamycin (0.5 µM). Primary hippocampal neurons and autapses were transduced with AAV2/9.iGluSnFR, AAV2/9.iGluSnFR-Y230F, AAV2/9.SnFR-γ2, AAV2/9.SnFR-γ8, AAV2/9.mScarlet-γ2 or AAV2/9.mScarlet-NETO2-γ2 at either early (DIV 1–3) or late (DIV 6) time points. Lentiviruses encoding homer1-tdTomato were added to primary neuronal cultures between 1 and 3 DIV.

## Patch-clamp fluorometry on HEK cells

Patch-clamp recordings of HEK cells co-expressing glutamate receptors and SnFR/SnFR-γ2/SnFR-γ8 were performed 2–3 days after transfection. Whole HEK293 cells were lifted into the outflow of a piezo-driven fast perfusion switcher for activation by glutamate. For whole-cell patch-clamp recordings, the external solution was composed as follows (in mM): 158 NaCl, 20 HEPES, 3 KCl, and 1 $CaCl_2$ (pH 7.4). The intracellular (pipette) solution contained (in mM): 135 KCl, 20 KF, 20 HEPES, 3 NaCl, 1 $MgCl_2$, and 2 Ethylene Glycol Tetraacetic Acid (EGTA) (pH 7.4). Pipettes had a resistance of 3–5 MΩ when filled with intracellular solution, and we used ISO-type pipette holders (G23 Instruments) to minimise pipette drift. After whole-cell configuration was obtained, cells were held at −50 mV and the currents were recorded using Axograph X (Axograph Scientific) via an Instrutech ITC-18 D-A interface (HEKA Elektronik). Excitation by a 488-nm diode laser (iChrome MLE, Toptica Photonics) for GFP was directed through a manual total internal reflection fluorescence (TIRF) input to an Olympus IX81 microscope. We used a ×40 Olympus objective (NA 0.6) for all recordings on HEK cells. Fluorescence intensities in response to 488-nm excitation were recorded sequentially with 20-ms exposure time, without binning, on a Prime 95B CMOS camera (Photometrics). Laser emission and camera exposure were triggered in hardware directly from the digitizer. Images were recorded with MicroManager (*Edelstein et al., 2014*) and analyzed with Fiji (*Schindelin et al., 2012*).

## Imaging on rat primary hippocampal neurons

Rat primary hippocampal cultures were imaged on DIV 16–18. The external solution for bath perfusion contained (in mM): 145 NaCl, 2.5 KCl, 10 HEPES, 2 $CaCl_2$, 1 $MgCl_2$, and 10 glucose (pH 7.3). Excitation by 488- and 561-nm lasers (iChrome MLE, Toptica Photonics) for GFP and Red Fluorescent Protein (RFP) was directed through a manual TIRF input to an Olympus IX81 microscope. We used a ×100 Olympus objective (UAPON 100x O TIRF, NA 1.49) for all recordings on bulk culture primary hippocampal neurons. Nonetheless, we did not image in TIRF mode because bleaching was too severe

in this condition. Fluorescence intensities in response to 488- and 561-nm excitation were recorded sequentially with 20-ms exposure time and 1 × 1 binning on a Prime 95B CMOS camera (1200x1200 pixels, 11 µm pixel size; Photometrics). GFP and RFP emission were split with a H 560 LPXR superflat beamsplitter (AHF, Germany), and recorded on the same frame after passing through ET525/50 m (GFP) and ET620/60 m (RFP) filters (both Chroma). Emission filters were mounted within an Optosplit II Bypass (Cairn Research). Exposures were timed to precede the −80 mV voltage steps. Images were recorded with MicroManager and analyzed with Fiji as described above.

## Hippocampal autaptic neuronal culture

Animal housing and use were in compliance with, and approved by, the Animal Welfare Committee of Charité Medical University and the Berlin State Government Agency for Health and Social Services (License T0220/09). Newborn C57BLJ6/N mice (P0–P2) of both sexes were used for all the experiments. Primary neurons were seeded on microisland astrocyte feeder layers that were cultured for 2 weeks before the neuronal culture preparation. Astrocytes derived from C57BL/6N mouse cortices (P0–P1) were plated on collagen/poly-D-lysine microislands made on agarose-coated coverslips using a custom-built rubber stamp to achieve uniform size (200 µm diameter).

Brains from wild-type (WT; C57/BL6N; P0-2) animals were removed and placed in 4°C cooled Hank's Buffered Salt Solution (HBSS; GIBCO Life Technologies, Germany). Hippocampi were carefully dissected out and placed in Neurobasal-A Medium supplemented with B27, Glutamax, (all from GIBCO Life Technologies), and penicillin/streptavidin (Roche, Germany; full-NBA) at 37°C in a heated shaker. Full-NBA was replaced with Dulbecco's modified Eagle medium (DMEM; GIBCO), supplemented with 1 mM $CaCl_2$ and 0.5 mM Ethylenediamine tetraacetic acid (EDTA; enzyme solution), containing papain (22.5 U/ml; CellSystems GmbH, Germany) and incubated for 45–60 min. The digestion was stopped by removing the enzyme solution and replacing it with an inactivating solution of DMEM supplemented with albumin (2.5 mg/ml) and trypsin inhibitor (2.5 mg/ml; both Sigma-Aldrich). The inactivating solution was removed after 5 min, and replaced with full-NBA. Tissue was dissociated mechanically and cells were counted on a Neubauer chamber.

Hippocampal neurons were seeded at $3 \times 10^3$ cells onto 30 mm coverslips previously covered with a dotted pattern of microislands of astrocytes for electrophysiological recordings in autaptic cultures, and at a density of $100 \times 10^3$ cells onto 30 mm coverslips previously covered with an astrocyte feeder layer for immunocytochemical staining. Neurons were then incubated at 37°C and 5% $CO_2$ for 12–18 days.

## Immunocytochemistry

In live labelling experiments to detect surface SnFR proteins, WT hippocampal neurons (DIV 12–16), infected with different iGluSnFR chimeric constructs, were incubated for 10 min at 37° C with an anti-GFP antibody. Neurons were rinsed with phosphate-buffered saline (PBS) and fixed in 4% (wt/vol) Paraformaldehyde (PFA)/4% sucrose, pH 7.4 for 10 min at room temperature, after which they were washed three times in PBS. After fixation, neurons were permeabilised in 0.3% Triton X-100, quenched in $NH_4Cl$ 50 mM, blocked in 2% BSA and incubated 1 hr at room temperature with mouse antibody against PSD95 (1:500; Thermo Science, MA1-046) and chicken antibody against MAP2 (1:2000; Chemicon, AB5543). Primary antibodies were labelled with Alexa Fluor 405 Affinipure donkey anti-chicken IgG (1:500; Jackson ImmunoResearch) and Goat anti-mouse Alexa Fluor 555 and goat anti-rabbit Alexa Fluor 647 (1:1000 each; Invitrogen).

In experiments to label VGlut puncta, at DIV 12–16, WT hippocampal neurons infected with the different iGluSnFR chimeric constructs were rinsed with PBS and fixed in 4% (wt/vol) PFA in PBS, pH 7.4 for 10 min at room temperature, after which they were washed three times in PBS. After fixation, neurons were permeabilised in PBS-Tween 20 (PBS-T), quenched in PBS-T containing glycine, blocked in PBS-T containing 5% normal donkey-serum and incubated overnight at 4°C with chicken monoclonal antibody against MAP2 (1:2000; Chemicon, AB5543) and guinea pig polyclonal antibody VGLUT 1 (1:4000; Synaptic System, 135304). Primary antibodies were labelled with Alexa Fluor 405 Affinipure donkey anti-chicken IgG and Alexa Fluor 647 Affinipure donkey anti-guinea pig IgG (1:500 each; Jackson ImmunoResearch).

Coverslips with the hippocampal cultures were mounted with Mowiol 4-88 antifade medium (Polysciences Europe). Neuronal images were acquired using an Olympus IX81 inverted epifluorescence

microscope at ×63 optical magnification with a CCD camera (Princeton MicroMax; Roper Scientific) and MetaMorph software (Molecular Devices).

At least three independent cultures were imaged and analyzed blind for each experiment. All images were acquired using equal exposure times and subjected to uniform background subtraction and optimal threshold adjustment. After background subtraction and threshold adjustment, images were converted to binary using FIJI. Raw values were exported to Prism 7 (GraphPad) for further analyses.

## Electrophysiology and imaging of autaptic cultures

Whole-cell voltage clamp recordings were performed on autaptic hippocampal neurons at DIV 14–18 at room temperature. Currents were acquired using a Multiclamp 700B amplifier and a Digidata 1440A digitizer (Molecular Devices). Series resistance was set at 70% and only cells with series resistances <12 MΩ were selected. Data were recorded using Clampex 10 software (Molecular Devices) at 10 kHz and filtered at 3 kHz. Borosilicate glass pipettes with a resistance around 3 MΩ were used and filled with an intracellular solution containing the following (in mM): 136 KCl, 17.8 HEPES, 1 EGTA, 4.6 $MgCl_2$, 4 $Na_2ATP$, 0.3 $Na_2GTP$, 12 creatine phosphate, and 50 U/ml phosphocreatine kinase; 300 mOsm; pH 7.4. Neurons were continuously perfused with standard extracellular solution including the following (in mM): 140 NaCl, 2.4 KCl, 10 HEPES, 10 glucose, 2 $CaCl_2$, 4 $MgCl_2$; 300 mOsm; pH 7.4. When using 4 mM $CaCl_2$, we reduced $MgCl_2$ to 2 mM, and in 0.5 mM $CaCl_2$, the $MgCl_2$ concentration was 5.5 mM. Changes in divalent concentration were achieved by local perfusion, and the evoked EPSC was used to monitor the change in response over a 2- to 3-min equilibration period. In experiments combining electrophysiology and imaging, movies were acquired using an Olympus IX81 inverted epifluorescence microscope at ×63 optical magnification, and 2x2 pixel binning (256 x 256 pixels field of view) with an Andor iXon EM + DU897E camera (16 µm pixel size)and iQ software. Spontaneous release was measured by recording mEPSC for 30 s at −70 mV and for an equal amount of time in 3 µM of the AMPA receptor antagonist NBQX to estimate false positives. We calculated the frequency as the difference in frequency between control and NBQX condition, and the amplitude was the difference of the amplitudes, weighted according to the normalised frequency difference. In some cases, the frequency and amplitude measured was similar for control and NBQX conditions (corresponding to few or no minis detected over background). We discarded amplitudes calculated where the frequencies (control vs. NBQX) differed by less than 0.1 Hz, where the amplitude in NBQX was not less than that in control or where the frequency of detected events were higher in NBQX. NBQX action was confirmed by the loss of the evoked EPSC. For each cell, data were filtered at 1 kHz and analyzed using template-based miniature event detection algorithms implemented in the AxoGraph X software. Action potential-evoked EPSCs were elicited by 2ms somatic depolarisation from −70 to 0 mV. Short-term plasticity was examined by evoking 5 action potentials with 200-ms interval (5 Hz). Data were analyzed offline using Axograph X (Axograph Scientific).

## Fluorescence data analysis

Background fluorescence signal was measured from 1 µm ROI with the smallest fluorescence intensity in each image that did not show any intensity change during recordings. The baseline level was subtracted from each recording.

For automatic selection of ROIs that responded to stimulation, and for extraction of fluorescence data for all ROIs, we wrote Fiji scripts that we include in supplementary files. For calculation of fluorescence change ($\Delta F/F$), the baseline fluorescence ($F_0$) was defined as the median fluorescence in the period before the response and then subtracted from $j$th acquisition frame in a set of $n$ frames and divided by $F_0$ to covert each trace to units of $\Delta F/F$ according to equation:

$$[\Delta F/F]_j = (F_j - F_o)/F_o, j = 1, 2...n$$

For short-term plasticity analysis, we designed a program written in PYTHON, SAFT (repository at https://github.com/agplested/SAFT; copy archived at swh:1:rev:c3f5615648978f9fdc82237c443e-d29eadb2fc15; *Plested, 2023*). The workflow of SAFT is shown in *Figure 9—figure supplement 2*. Briefly, the baseline was interactively subtracted using the asymmetric least squares smoothing algorithm (*Eilers and Boelens, 2005*). Peak finding was done using the SciPy-wavelet transform algorithm (*Virtanen et al., 2020*) and manually curated by the user in the graphical interface. Peak locations

from the mean waveform of responses from all identified ROIs was used to extract peak responses from all ROIs. ROIs with low signal to noise or rundown were excluded, where noted. For coefficient of variance analysis, we fit the following formula (*Faber and Korn, 1991*):

$$CV = \frac{\sigma}{\mu} = \left[ \frac{(1 - P_R)}{n \cdot P_R} \right]^{\frac{1}{2}}$$

where $\sigma$ is the standard deviation, $\mu$ is the mean response, and $n$ is the number of release sites. A global fit using the Solver in Excel was performed, allowing a different release probability ($P_R$) for each calcium condition.

To fit the peak response histograms from each ROI, we used a mixed Gaussian model for fitting the width and number of release sites needed. We first made a histogram from pooled responses, and fitted it with a multiple Gaussian function to get an estimate of the quantal size which was used for subsequent fits. Our peak detection algorithm assigned failures the value 0 by design, because this gave a more stable and reliable output. One disadvantage was that no symmetric peak around zero was created from failures. All failures had value 0 and were collected in the first bin of the histogram. For global fits across conditions, we used OPTIMISE SciPy (*Virtanen et al., 2020*) on a flattened array of peaks amplitudes, assuming that $N$ and $q$ were constant over calcium variation but release was not and depended on release probability ($P_R$, binomial model) or release rate ($\lambda$, Poisson model). In the binomial model, the amplitudes of each Gaussian component were determined with the binomial probability mass function (SciPy) according to the release probability, $P_R$, and a common scale factor both of which were optimised to obtain the fit. For the Poisson model the amplitudes of each Gaussian component were determined with the Poisson probability mass function (SciPy) according to the release rate, $\lambda$, and a common scale factor both of which were optimised to obtain the fit. For the Poisson fit, the widths of each Gaussian component increased in proportion. For each fit, the Kolmogorov–Smirnov test was used to determine goodness of fit. We used the KSTEST function in SciPy (*Virtanen et al., 2020*) function and determined the necessary cumulative distribution by hand. Best $N$ values were determined by brute force based on the K–S value and checked manually.

## Acknowledgements

We thank Marcus Wietstruk for molecular biology, Ljudmila Katchan for developing the N-terminal extracellular TARP insertion site, Thorsten Trimbuch and the Viral Core Facility of the Charité for virus production, Heike Lerch, Kordelia Hummel, and Niccolò Pampaloni for assistance with neuronal cell culture and Berit Söhl-Kielczynski for help with immunocytochemistry. We thank Teresa Giraldez, Marina Mikhaylova, and Melissa Herman for comments on the manuscript. This work was supported by the ERC grant 647895 'GluActive' (to AJRP), the Deutsche Forschungsgemeinschaft (DFG, German Research Foundation) under Germany's Excellence Strategy—EXC-2049-390688087 (to both CR and AJRP) and Heisenberg Professorship to AJRP (project numbers: 323514590 and 446182550).

## Additional information

### Funding

| Funder | Grant reference number | Author |
| --- | --- | --- |
| Deutsche Forschungsgemeinschaft | 390688087 | Andrew JR Plested Christian Rosenmund |
| European Research Council | 647895 | Andrew JR Plested |
| Deutsche Forschungsgemeinschaft | 323514590 | Andrew JR Plested |
| Deutsche Forschungsgemeinschaft | 446182550 | Andrew JR Plested |

| Funder | Grant reference number | Author |
|--------|------------------------|--------|

The funders had no role in study design, data collection, and interpretation, or the decision to submit the work for publication.

## Author contributions

Yuchen Hao, Data curation, Formal analysis, Investigation, Methodology, Writing – original draft, Writing – review and editing; Estelle Toulmé, Conceptualization, Data curation, Formal analysis, Supervision, Investigation, Visualization, Methodology, Writing – original draft, Writing – review and editing; Benjamin König, Software, Formal analysis; Christian Rosenmund, Conceptualization, Resources, Supervision, Funding acquisition, Project administration; Andrew JR Plested, Conceptualization, Resources, Data curation, Software, Formal analysis, Supervision, Funding acquisition, Validation, Investigation, Visualization, Methodology, Writing – original draft, Project administration, Writing – review and editing

## Author ORCIDs

Yuchen Hao (iD) http://orcid.org/0000-0002-0042-6576
Estelle Toulmé (iD) http://orcid.org/0000-0001-8734-3484
Christian Rosenmund (iD) http://orcid.org/0000-0002-3905-2444
Andrew JR Plested (iD) http://orcid.org/0000-0001-6062-0832

## Ethics

Animal housing and use were in compliance with, and approved by, the Animal Welfare Committee of Charité Medical University and the Berlin State Government Agency for Health and Social Services (Licenses T0220/09 and FMP_T 03/20). Newborn C57BLJ6/N mice (P0–P2) and rats (P1–P3) of both sexes were used for all the experiments.

## Decision letter and Author response

Decision letter https://doi.org/10.7554/eLife.84029.sa1
Author response https://doi.org/10.7554/eLife.84029.sa2

# Additional files

## Supplementary files

- MDAR checklist
- Source code 1. imageJ macro to find regions of interest.

## Data availability

Custom software is available at https://github.com/agplested/SAFT (copy archived at swh:1:rev:c3f5615648978f9fdc82237c443ed29eadb2fc15).

The following dataset was generated:

| Author(s) | Year | Dataset title | Dataset URL | Database and Identifier |
|-----------|------|---------------|-------------|-------------------------|
| Hao Y, Toulmé E, König B, Rosenmund C, Plested A | 2023 | Targeted sensors for glutamatergic neurotransmission | https://zenodo.org/record/7512561 | Zenodo, 10.5281/zenodo.7512561 |

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
