## [Editor Report]

This manuscript addresses the potential value of a "tagged" version of iGluSnFRs with the idea that this approach provides a more localized measure of glutamate release at synapses. Although the new sensor does not have an increase in signal-to-noise ratio, the authors nicely address the potential advantages and limitations of their sensor and the experiments provide an important test of the localized expression of such a sensor.

---

## [Decision Letter]

**Decision letter after peer review:**

[Editors’ note: the authors submitted for reconsideration following the decision after peer review. What follows is the decision letter after the first round of review.]

Thank you for submitting your work entitled "Targeted sensors for glutamatergic neurotransmission" for consideration by *eLife*. Your article has been reviewed by 3 peer reviewers, and the evaluation has been overseen by a myself as Senior Editor. The following individuals involved in review of your submission have agreed to reveal their identity: Kevin J Bender (Reviewer #3).

Comments to the Authors:

We are sorry to report that, after a joint discussion with the reviewers, we have decided that your work will not be considered further for publication by *eLife*. The reviewers understood the potential value of the approach to develop a synaptically targeted version of iGluSnFR, but raised a number of technical and conceptual issues that we think would require more than modest revisions to support the conclusions.

*Reviewer #1 (Recommendations for the authors):*

Hao et al. targeted the genetically encoded glutamate sensor iGluSnFR to synapses by fusion with auxiliary subunits of the AMPA receptor, y2 (stargazin) and y8. They report decreased affinity and increased response stability compared to non-anchored iGluSnFR. Early infection with SnFR-γ2 (or just stargazin) blocked EPSCs, suggesting that AMPARs were displaced from their postsynaptic sites. This effect was less extreme when cultures were transfected late (DIV6), but currents were still down by 50% compared to iGluSnFR-transfected neurons, while presynaptic function appeared to be normal.

To analyze the imaging data, the authors developed a strategy to use the signal (stimulation-evoked increase in fluorescence) to select their regions of interest. This allowed them to identify sites of glutamate release, but is not an unbiased sampling of the synaptic population. The authors show only one example of colocalization with homer (Figure 3a), leaving some doubt as to what fraction of indicator molecules was successfully targeted. For a well-targeted indicator, it should be possible to use resting fluorescence spots to select ROIs.

During repeated stimulation at 5 Hz in 2 ca^2+^, SnFR-y2 produced stable responses while iGluSnFR responses decreased. EPSCs in SnFR-y2 neurons were smaller (Figure 4C, note split axis) but displayed similar short-term plasticity (Figure 7C). Comparing neurons, SnFR-y2 was highly correlated with short term facilitation / depression while iGluSnFR signals were not. The authors speculate that the poor correlation of iGluSnFR is due to run-down, but this would manifest as exaggerated depression, not spurious facilitation as the data suggest. So the reason for the improved performance of SnFR-y2 is not entirely clear. The authors then use the SnFR signals to analyze single synapses in autaptic culture. They show very nicely that the glutamate output is a function of extracelluar ca^2+^, providing direct proof for multivesicular release at individual synapses. Full optical quantal analysis requires measurable responses to the release of a single quantum, which SnFR-γ2 and SnFR-γ8 do not seem to provide (Note that the spontaneous fluorescence transients recorded without TTX (Figure 2) are potentially multivesicular events).

In summary, the improvement of the new variants compared to iGluSnFR could be due to their decreased affinity for glutamate, resulting in selection of the strongest synapses and very focal signals, but lack of sensitivity to the fusion of a single vesicle. The price to pay for synaptic targeting, strong alteration of postsynaptic receptor composition, seems relatively high and may prevent widespread adoption of the new variants.

Concerns:

1) Synaptic targeting: The appearance of the targeted indicator is punctate, but the ROIs that were selected by their signal often have no higher resting fluorescence than their surroundings (Figure 2) while the brightest spots apparently produce no signal. By co-expressing homer, the authors tried to quantify colocalization, but show only one single SnFR-y2 spot (n=1) that is colocalized with homer. (As a side remark, iGluSnFR SIGNALS should also be colocalized with homer, but the example is not). In the text, they state that the complexity of expression (?) precluded colocalization analysis. Thus, in spite of the author's efforts, evidence for successful synaptic tagging is lacking, and the schema presented in Figure 9 (88% of SnFR-y2 inside PSD) seems optimistic. All differences in y2-sensor responses compared to iGluSnFR (e.g. spatially restricted responses) could be due to its reduced affinity for glutamate and in consequence, selection bias towards the most powerful synapses.

2) Run-down of iGluSnFR vs. stability of SnFR-y8 (Figure 6): As SnFR-y8 has a lower affinity for glutamate than iGluSnFR, the ROI detection algorithm finds fewer active pixels on SnFR-y8 neurons. I would expect these to correspond to the strongest and most reliable sources of glutamate. iGluSnFR with its higher affinity will also pick up synapses with a small pool of release-ready vesicles. Small synapses are likely to display pronounced depression during a train. So, the stability of SnFR-y8 responses might reflect a selection bias for the strongest synapses. Figure 6g does little to rule out this possibility, since the size of the compound ROI does not reflect the strength of the synapse (but may pool the response of several synapses). The interpretation of the authors, that synaptic targeting somehow stabilized the fluorescence of the SnFR, I find much less likely. Another scenario that cannot be ruled out is stronger illumination of the iGluSnFR cells (perhaps due to low expression levels), resulting in increased photobleaching during the stimulation train. To monitor indicator bleaching, the resting fluorescence (F0) during the train should be reported for both indicators. In figure 6e, fluctuations in the time course are highly correlated across ROIs, which points to an artifact (non-stationary stimulation or illumination). To solve the mystery of run-down, it would be very helpful to see the EPSCs generated by this stimulation in iGluSnFR and in SnFR-y8 neurons.

3) Quantal analysis: As the biological meaning of the quantal parameters is best defined in single synapses, I will first comment on the single synapse experiments (Figure 8, Figure S6, Figure S8). The authors analyze single ROIs at three different calcium concentrations, resulting in increasing response amplitudes. This is interpreted as multi-vesicular release from a single synapse. While I follow this general interpretation, it seems the sensitivity of sensor/imaging system is not sufficient to detect the release of individual vesicles. In 0.5 mM ca^2+^, the recorded SnFR-y8 traces show no difference between simulated and non-stimulated epochs, strongly suggesting that the fluctuations in fluorescence intensity are noise. Consequently, the histograms of 0.5 mM ca^2+^ responses show no separation between failures and sucesses, just a single peak. To assume that this peak corresponds to the release of a single vesicle, the quantal response (q), is wrong. The example traces from iGluSnFR at 0.5 mM ca^2+^ look slightly more promising (Figure S6), but again, the histogram does not show a separation between failures and successes, and the compound histogram treats the entire first peak as the quantal response (i.e., assumes there were no failures in 0.5 mM ca^2+^). A method that lacks the sensitivity to detect single vesicle fusion is not useful for quantal analysis: Due to increasing variance and indicator saturation, the separation between multi-vesicular events will be less clear, not better, than the separation between uni-vesicular events and failures.

4) Quantal analysis of autaptic connections (Figure 7): The authors use the CV method developed for paired recordings, which seems appropriate for an autaptic neuron. Compared to the analysis of single synapses, the meaning of N is less well defined here. Early studies assumed that one synapse can only release one vesicle at a time, which makes N the number of connecting synapses and Pr the synaptic release probability. For autaptic cultures, multi-vesicular release is established, so N is the number of "release sites", several of which can be located in a single synapse. Thus, N must be larger than the number of recorded synapses, which in this case is the number of ROIs.

It is very difficult to understand Figure 7. Trains of 5 AP are evoked at 5 Hz ten times in 3 different Ca concentrations and recorded in an unspecified number of ROIs (synapses). In panel c, facilitation/depression during the train is compared for electrical and optical recordings. I assume these points (n=6) correspond to different cells, not different ROIs (please indicate the point corresponding to the cell analyzed in panels a and b). SnFR-y2 responses are better correlated with the electrical responses in 2 and 4 Ca, but SnFR-y2 synapses tend towards depression compared to iGluSnFR synapses. This population difference in EPSC dynamics raises the nasty possibility of presynaptic effects caused by SnFR-y2 expression (but due to low sample size, may just be a fluke).

In Figure 7D, mean and standard deviations are calculated, but the basis is not clear. I am assuming each ROI produces 5 points in panel D, each averaged over the 10 repeats. It would be helpful to explain this more clearly. The CVs (slopes in panel D) are related to N and Pr. As the authors state, they end up with 3 numbers for an equation system with 4 variables. It is not clear to me how fitting can help in this situation (as there should be no error, regardless which number is chosen for N). So perhaps panel F shows the numbers they chose for N, perhaps based on the number of ROIs? (The legend of F is cryptic. Please indicate in E and F which points correspond to the cell shown in D.) This would be based on the assumption of univesicular synapses, though, which they refute in Figure 8. In any case, they point out that this N is similar to the N reported by Bekkers and Stevens, who were interested in the number of release sites connecting two neurons. The numbers (N = 4-7) seem very low for a connection producing 2 nA EPSCs (400 pA per uniquantal synapse?). It could of course be argued that the autaptic connection is made of many more synapses than the few that are sampled optically. The N determined by optical quantal analysis, however, has to be higher than the number of ROIs (active synapses), it cannot be lower. If the N is assumed to be equal to the number of ROIs (uniquantal synapses), it makes no sense to call the process quantal analysis (as in this case, the distribution of N values (Figure 7F) is not the result of any fitting procedure). Without understanding the basis of N, I cannot interpret the meaning of Pr (synaptic? vesicular?).

Suggested improvements:

Provide conclusive evidence for successful synaptic targeting, e.g. by homer/y2 co-transfection of individual neurons and high-resolution imaging. Targeting is expected to be most specific at low expression levels.

I am not sure how to improve the quantal analysis (more excitation light?). A symmetrical failure peak should appear exactly at df/f = 0 which is absent from all histograms. Perhaps the problem is background subtraction (leading to division by zero) or the handling of negative df/f values. If there is no way to separate failures from low probability release events in individual trials, I will remain skeptical about multi-peaked compound histograms apparently separating 4 from 3 simultaneously released vesicles.

Please commit to a clear definition / interpretation of the extracted parameters from the coefficient of variance method, specifically in relation to the number of ROIs. At present, the biological interpretation of N and Pr with respect to autaptic synapses is unclear, at least to me.

*Reviewer #2 (Recommendations for the authors):*

1. According to the authors, one principal advantage of their approach is that SnFR-y2/8 provides a more 'spatially precise' signal compared with iGluSnFR. Clearly, an optical sensor that is proposed as expressed within a nanoscopic membrane domain will yield a more spatially constrained signal compared with the sensor expressed evenly over cell membranes. However, this simply reflects the sensor distribution properties rather than anything else. In words, SnFR-y2/8 will tend to report the brightest glutamate signal where SnFR-y2/8 is accumulated, rather than where the glutamate concentration is highest (e.g., release site proximity). In contrast, the sensor homogenously distributed in space, such as the original iGluSnFR variants, should provide unbiased readout of glutamate hotspots. It appears therefore that the authors strategy is somewhat self-defeating.

2. The authors use TIRF imaging, and single-line diode lasers as an excitation source. This type of imaging is only suitable for monolayer cultures: whether their sensor will be efficient when imaged in organized brain tissue using two-photon excitation is not clear. The claim on a methodological advance appears therefore premature.

3. The other key statement is that iGluSnFR is prone to photobleaching (rundown) more than is SnFR-y2/8. First, fluorophore's photobleaching properties in 1P as opposed to 2P mode could be very different, which has not been addressed or explored. Second, this observation is surprising because several recent studies have documented a fairly stable iGluSnFR signal over multiple cycles of glutamate release imaged at 'quantal' resolution, both in 1P and 2P excitation regimes, both in cultures and in acute slices (e.g., Tagliatti et al., 2020 PNAS 117: 3819; Jensen et al. 2019 Nat Commun 10: 1414). The authors do not seem familiar with these studies. Dye photobleaching depends on multiple imaging parameters starting with laser power: this has not been investigated consistently in the present work.

4. Stargazin overexpression has been used as a principal tool for the iGluSnFR targeting to synapses, but the authors report that this renders a proportion of synapses nonresponsive to glutamate (Figure 4). That the method interferes with the physiological integrity of synaptic circuits, or at least requires some additional experiment-specific manipulations to minimize it, does not speak in its favor.

5. The signal-to-noise ratio of the SnFR-y2 signal does not appear improved compared with that of iGluSnFR (Figure 8A).

6. The quantal-analysis histograms presented here (Figure 8C-D) appear noisier hence less reliable than similar or related analyses in the aforementioned publications that employed iGluSnFR.

*Reviewer #3 (Recommendations for the authors):*

Hao and colleagues developed new variants of the glutamate sensor iGluSnFR, termed SnFR-γ2 and SnFR-γ8, that are fusion proteins with postsynaptic density (PSD) proteins Stargazin and γ-8. This chimeric protein thus localizes specifically at the PSD. These new variants are characterized, using heterologous expression systems and hippocampal neuron microisland cultures, allowing one to monitor autapses with simultaneous electrical and optical access. Overall, SnFR-γ2 outperforms traditional iGluSnFR in terms of signal localization; presumed single synapses are observed with limited "spillover" of signal to neighboring regions, and imaged transients are amenable to traditional noise analysis. Some concern regarding competition for PSD membrane is raised due to overexpression of these variants, but it appears that such overexpression artifacts can be avoided by ensuring that SnFR-γ2 is delivered after synapses are largely formed. This could be an important tool for the field, as it improves one's ability to resolve the activity of single glutamatergic synapses with sufficient signal to noise. Work here has shown feasibility in cultures systems. Future work will need to show similar performance levels in acute slice and in vivo, though based on past observations with iGluSnFR, this performance is likely within reach.

The main question I have is one of extensibility: can this approach work in more intact systems, or even in vivo? I recognize that asking such a question involves an entirely new dataset, and am not proposing that the authors engage in such an effort after already doing an excellent job characterizing these GluSnFR variants. Rather, I'd hope that the authors would expand on their discussions, which is largely focused on questions of release dynamics observed with their sensors, to also include potential technical advantages or limitations of these variants in other preparations.

Comments below are aimed at improving interpretation of data reported herein:

1) A control for infection is needed for autapse data. Please make parallel recordings in cells infected with control viruses that lack any glutamate sensor to determine if these currents are in the normal range at these ages.

2) Data in Figure 5C and F should be analyzed quantitatively by calculating correlation coefficients for each pair of data. If there are differences in the relative separation of each region (I understand that these were chosen by hand) then a potential comparison could be made by plotting correlation coefficients vs. centroid distance of paired ROIs.

3) Data in Figure 7C, 0.5 mM. There is an obvious outlier near 7 P5/P1 for the SnFR-gamma2. This is likely due to a very low P1 value for this one cell, indicative of excess failures. I'd be curious to know if any correlation holds if this one datapoint is held out of the dataset. If so, perhaps the authors could explain this observation in the main text.

4) Figure 7, data related to optical quantal analysis. Considering that these are autaptic recordings, with superb electrical access, one should be able to perform traditional electrophysiological quantal analysis and determine whether SnFR is allowing for identification of all synapses. This is a critical analysis that should be made on a cell-by-cell basis, paired with optical analyses; however, if this is not feasible at this time, some information could be gleaned from separate recordings, given that you observed fairly consistent numbers of sites optically (4 to 7). This would address your concern that the detection methods bias towards high Pr synapses. Though, if there are more synapses made a significant distance from the coverslip, then they would never be imaged under TIRF microscopy, obviating this request.

[Editors’ note: further revisions were suggested prior to acceptance, as described below.]

Thank you for resubmitting your work entitled "Targeted sensors for glutamatergic neurotransmission" for further consideration by *eLife*. Your revised article has been evaluated by Gary Westbrook (Senior Editor) and a Reviewing Editor.

The manuscript has been improved but there are some remaining issues that need to be addressed, as outlined below:

This manuscript is an attempt towards synapse specificity for glutamate probes. The current probe's utility is limited to specific cases, and it has what could be considered advantages and drawbacks, even in that specific case (CMOS imaging of cultured cells). It is commendable how carefully the authors characterized side effects of this sensor on synaptic function. The reviewers agreed that several issues require clarifications in the text.

1. Please explain that any glutamate rises or waves in the tissue, any significant glutamate spillover signaling, or even fluctuations in tissue optical conditions, could be falsely perceived as local synaptic signals by an PSD-constrained sensor rather than by an evenly distributed sensor. This is a key limitation of the current method.

2. The authors seem to insist that the use of iGluSnFR or any sensor labeling is generally disadvantageous. It is advantageous in most cases. In terms of this sensor's fluorescence properties, there is no evidence for improved performance in the manuscript.

3. Please consider the specific points raised by the reviewers and address them in the text to the extent possible.

*Reviewer #1 (Recommendations for the authors):*

The authors have provided their explanations and rebuttal regarding the previous comments, albeit without new experimental evidence.

1. It appears that the authors did not fully understand the main objection. Or perhaps it was not explained clearly enough. To reiterate: SnFR-y2/8 expressed locally at the synapse may in fact sense and report glutamate that is released elsewhere in the vicinity, thus giving a false impression of its local synaptic release. In other words, SnFR-y2/8 may report the spillover signal as well as the actual synaptic signal.

It is true that, in 2PE mode, in organized brain tissue, iGluSnFR will report optical signal integrated, due to diffraction of light, within the PSF depth of ~800 nm, etc., and thus may include glutamate release events, if any, from nearest synaptic neighbors. However, the same will happen if such neighbors express SnFR-y2/8: their optical signal will still be integrated within the diffraction-limited PSF in 2PE imaging mode.

In the case when no glutamate spillover ever occurs in the preparation of interest, both iGluSnFR and SnFr-y2/8 will report true synaptic signals when such happen. In this case, however, glutamate rises or waves arising from other sources, such as astroglia, will still be reported by SnFr-y2/8 very locally, giving an impression of local synaptic events – whereas iGluSnFR will report the entire glutamate 'landscape'. Thus, SnFr-y2/8 does not seem to have any principal spatial-resolution advantage over iGluSnFR.

2. Authors' attention is drawn to yet another recent publication (Mendonca et al. 2022 Nat Commun 3497) showing an excellent stability, S/N ratio, etc. for highly localized glutamate release events detected with iGluSnFR. That this sensor is performing not as satisfactory in the present authors' hands cannot be a basis for their extrapolated claim.

3. The authors concede that the S/N ratio of their probe readout is probably no better than that of iGluSnFR.

4. The authors have made no attempt to check their probe performance in 2PE mode and/ or in organized brain tissue.

5. The amplitude histograms shown in Figure 9C-D do not match satisfactorily the best-fit quantal analysis curves. While the authors acknowledge the difficulty, the reasons for displaying unsatisfactory quantal analysis data are not clear.

6. The entire concept would have made a much greater impact if the authors aimed to express the sensor at a specific, functionally or genetically distinct, sub-population of synapses.

*Reviewer #2 (Recommendations for the authors):*

The authors have addressed concerns raised by all three reviewers to the best of their ability. Their commentary that untagged SnFR is likely detecting large hotspots of both synaptic and spillover is taken well. Though the concern regarding how well the tagged variant works remains in question. If one can only resolve a few synapses (less than 10 per cell were analyzed) of what the authors suggest are ~100 potential autapses, are these few ROIs representative of the whole? This may be a good point of discussion.

Given the way in which *eLife* is modifying its review process and overall publishing criteria, this work should be accepted. It represents a new tool that is well characterized. Advantages and flaws are discussed well.

*Reviewer #3 (Recommendations for the authors):*

In their revised version, the authors address questions about synaptic localization of their GEGI by demonstrating good overlap with the PSD95 signal (new Figure 3). The explanation of Figure 8 has been much improved, quantal parameters are now well defined. The correlation between electrical and optically measured depression during a train (Figure 8c) is indeed much better for their targeted indicator compared to iGluSnFR, which is a strong argument that synaptic (and not extrasynaptic) glutamate is reliably measured by their sensor. In this context, it is an advantage of the autaptic model that all synapses on a given neuron have the identical history of activity and therefore express similar (but cell-specific) short-term plasticity. I do not like the analysis of rundown in Figure 7F: Splitting a continuous distribution into 2 groups, using an arbitrary threshold, then counting the number of cases (ROIs) on either side of the threshold, is not good statistical practice. The analysis in Figure S5 I like much better, it shows no significant difference between the two indicators. As the authors cannot offer a convincing mechanistic explanation why a difference in rundown would be expected, I suggest downplaying this point (getting rid of Figure 7E-G, sticking with the message of S5).

Apart from this quibble, the science is sound and well presented. The separation of quantal histogram peaks is impressive and certainly aided by localizing the indicator to the places of highest glutamate concentration. Extrasynaptic indicator molecules, exposed to a near-continuous range of glutamate concentrations, would be expected to widen the peaks considerably (as shown in the iGluSnFR example in Figure 9c). For future tool development and targeting efforts, the technical information and precise measurements are very useful, even as a somewhat cautionary tale with regard to potentially severe side effects of tool expression.

---

## [Author Response]

[Editors’ note: the authors resubmitted a revised version of the paper for consideration. What follows is the authors’ response to the first round of review.]

Reviewer #1 (Recommendations for the authors):Hao et al. targeted the genetically encoded glutamate sensor iGluSnFR to synapses by fusion with auxiliary subunits of the AMPA receptor, y2 (stargazin) and y8. They report decreased affinity and increased response stability compared to non-anchored iGluSnFR. Early infection with SnFR-γ2 (or just stargazin) blocked EPSCs, suggesting that AMPARs were displaced from their postsynaptic sites. This effect was less extreme when cultures were transfected late (DIV6), but currents were still down by 50% compared to iGluSnFR-transfected neurons, while presynaptic function appeared to be normal.

We disagree with this otherwise good summary in only one respect: we do not claim any change in affinity. We apologise that we did not underline that a change of apparent glutamate affinity by a factor of 2 is not meaningful. We now added text near line 153 to emphasize that this is a slight change, compared to other published variants.

To analyze the imaging data, the authors developed a strategy to use the signal (stimulation-evoked increase in fluorescence) to select their regions of interest. This allowed them to identify sites of glutamate release, but is not an unbiased sampling of the synaptic population. The authors show only one example of colocalization with homer (Figure 3a), leaving some doubt as to what fraction of indicator molecules was successfully targeted. For a well-targeted indicator, it should be possible to use resting fluorescence spots to select ROIs.

Thank you. We take this criticism seriously. Originally, we worked hard to get a good measure of colocalization from the sensor itself in live cell imaging but, as the referee identified, this was not possible. We now include new Figure 3, where we measure the localization of SnFR variants by immunofluorescence microscopy on live-labelled, fixed cells. These experiments show the enrichment at synaptic sites for SnFR-g2. However, the enrichment is not absolute, presumably because sensors are not anchored at the PSD but rather exchange during an experiment. The mild enrichment perhaps relates to the somewhat paradoxical observation that we do not have major bleaching despite tethering.

During repeated stimulation at 5 Hz in 2 ca^2+^, SnFR-y2 produced stable responses while iGluSnFR responses decreased. EPSCs in SnFR-y2 neurons were smaller (Figure 4C, note split axis) but displayed similar short-term plasticity (Figure 7C). Comparing neurons, SnFR-y2 was highly correlated with short term facilitation / depression while iGluSnFR signals were not. The authors speculate that the poor correlation of iGluSnFR is due to run-down, but this would manifest as exaggerated depression, not spurious facilitation as the data suggest. So the reason for the improved performance of SnFR-y2 is not entirely clear. The authors then use the SnFR signals to analyze single synapses in autaptic culture. They show very nicely that the glutamate output is a function of extracelluar ca^2+^, providing direct proof for multivesicular release at individual synapses. Full optical quantal analysis requires measurable responses to the release of a single quantum, which SnFR-γ2 and SnFR-γ8 do not seem to provide (Note that the spontaneous fluorescence transients recorded without TTX (Figure 2) are potentially multivesicular events).

During repeated stimulation, some iGluSnFR responses ran down dramatically, and some did not. But the advantage of SnFR-Gamma2 is that almost no responses ran down. So there is no need to select between them.

We do not agree that rundown would manifest as depression (STD). Rundown (the overall loss of signal) is happening not during the train (that would be depression) but instead to all the responses of certain ROIs as the experiment progresses (see now Figure 7). This effect tends to make the later responses closer to the noise and therefore apparently more random than the earlier ones. This effect does not happen with our sensor (Figure 7).

For full optical quantal analysis, in the ideal case, single responses can be well resolved. The reviewer is right, our sensors are not ideal in this regard. However, we feel that idea that this is absolutely required is challenged by our excellent estimation of quantal size from global fitting at different calcium concentrations.

We now added a note to point out that the poor resolution of the single quanta is a drawback (near line 478).

In summary, the improvement of the new variants compared to iGluSnFR could be due to their decreased affinity for glutamate, resulting in selection of the strongest synapses and very focal signals, but lack of sensitivity to the fusion of a single vesicle. The price to pay for synaptic targeting, strong alteration of postsynaptic receptor composition, seems relatively high and may prevent widespread adoption of the new variants.

Briefly, no, the affinity is not changed enough for this to be the case. We measured this in close to native conditions (mammalian cells, not in a cuvette) in Figure 1. We did not particularly expect widespread adoption of these variants (a few labs have requested them from Addgene).

What we show is the principle of a massive quantitative improvement (Figure 6 and 7) from targeting, and this should be taken seriously. The performance of our sensor is somewhat different to the prevailing view that tethering the sensor should lead to bleaching. These are important observations for the field in general.

During our process of revising this manuscript, the team at Janelia have developed their new best in class indicator, including the Stg-C-terminus to localize it. It seems this works very well in their hands too but is not so bright in vivo. They recommend it for in vitro work. This development follows exactly from what we present in this manuscript, and their data complement ours. It seems the principle will have good adoption. We are very glad that open science (we put our MS on Biorxiv and gave the plasmids and vectors away) helping to promote innovation in this way.

Concerns:1) Synaptic targeting: The appearance of the targeted indicator is punctate, but the ROIs that were selected by their signal often have no higher resting fluorescence than their surroundings (Figure 2) while the brightest spots apparently produce no signal. By co-expressing homer, the authors tried to quantify colocalization, but show only one single SnFR-y2 spot (n=1) that is colocalized with homer. (As a side remark, iGluSnFR SIGNALS should also be colocalized with homer, but the example is not). In the text, they state that the complexity of expression (?) precluded colocalization analysis. Thus, in spite of the author's efforts, evidence for successful synaptic tagging is lacking, and the schema presented in Figure 9 (88% of SnFR-y2 inside PSD) seems optimistic. All differences in y2-sensor responses compared to iGluSnFR (e.g. spatially restricted responses) could be due to its reduced affinity for glutamate and in consequence, selection bias towards the most powerful synapses.

We agree, our original analysis of colocalization with Homer in live cells was not convincing. We think this is because of a large amount of ER labelling by all membrane-bound constructs. We now point this out directly in the text near to line 201. We have addressed this concern with a new Figure 3 where we show an enrichment of surface SnFR-g2 (using a GFP antibody) and SnFR-g8 against PSD95 sites using immunohistochemistry.

To the point about iGluSnFR signal and colocalization: the Homer signal trivially corresponds to a synapse where there was no release in this part of the movie.

We take the point that the sketch (now) Figure 10 might appear that we are claiming something optimistic, but we do not mean imply any ratio of 88%. The ratio of extrasynaptic membrane to synaptic membrane is massive, and we do not depict this to scale, nor do we claim to. We added a note to the legend to make this clear.

We think that a really important result in our paper is that by using a non-targeted high-affinity sensor (iGluSnFR), you may unfortunately sample glutamate that is irrelevant for synaptic responses. This is consistent with the recent preprint from Janelia that provides evidence that iGluSnFR is excluded from synapses, and prefers extrasynaptic glutamate. All the successful work so far has used presynaptic anatomy to restrict the spatial aspect of expression, or some kind of filtering in the analysis to restrict it to “good” spots. We show that many of the responses of iGluSnFR to evoked release are massive in area, making their relation to synaptic glutamate implausible. This fact has been overlooked or ignored in previous work that concentrated on a presynaptic locus.

Finally, there is now evidence (from the Janelia Farm preprint) that the Stg C-terminus (inspired by our work) gives the best synaptic localization of iGluSnFR.

2) Run-down of iGluSnFR vs. stability of SnFR-y8 (Figure 6): As SnFR-y8 has a lower affinity for glutamate than iGluSnFR,

The affinity of the three sensors (iGluSnFR, SnFR-g2, SnFR-g8) is essentially the same.

The ROI detection algorithm finds fewer active pixels on SnFR-y8 neurons. I would expect these to correspond to the strongest and most reliable sources of glutamate. iGluSnFR with its higher affinity will also pick up synapses with a small pool of release-ready vesicles. Small synapses are likely to display pronounced depression during a train. So, the stability of SnFR-y8 responses might reflect a selection bias for the strongest synapses. Figure 6g does little to rule out this possibility, since the size of the compound ROI does not reflect the strength of the synapse (but may pool the response of several synapses). The interpretation of the authors, that synaptic targeting somehow stabilized the fluorescence of the SnFR, I find much less likely. Another scenario that cannot be ruled out is stronger illumination of the iGluSnFR cells (perhaps due to low expression levels), resulting in increased photobleaching during the stimulation train. To monitor indicator bleaching, the resting fluorescence (F0) during the train should be reported for both indicators. In figure 6e, fluctuations in the time course are highly correlated across ROIs, which points to an artifact (non-stationary stimulation or illumination). To solve the mystery of run-down, it would be very helpful to see the EPSCs generated by this stimulation in iGluSnFR and in SnFR-y8 neurons.

We are grateful for this careful critique and the excellent ideas for giving more insight into what is going on. The expression levels are similar between the different constructs, although the baseline fluorescence was generally a bit less for the SnFR-g2 constructs. The illumination was not changed between different experiments, which were always done on the same day. We now note this (near line 296), and we made a new supplementary figure S5 to look show all of these data, to answer this point. We make comparisons between iGluSnFR and SnFR- g2, not SnFR-g8 because few cells in this condition gave enough ROIs for this comparison to make sense.

We now report in new Supplementary Figure S5 the baseline fluorescence (F0) for the cells where we also recorded EPSCs. We show exemplary traces for the cells that we analyse in detail in other sections, and summarise the (lack of) rundown for the F0 signal across all the cells. The baseline fluorescence was less for SnFR-g2. We never mentioned bleaching in our original submission and we now explicitly point out that there is no substantial bleaching (near line 311). We also made the comparison that the referee suggests between EPSC rundown and fluorescence rundown. We describe it near line 320.

The observation of the correlation of peaks across ROIs is a fascinating prospect. It is much clearer for iGluSnFR than for SnFR. The strong correlation that the reviewer sees in now-Figure 7E is related to the phenomenon shown in now-figure 6 – the responses of iGluSnFR at neighbouring sites are highly correlated. Large ROIs are also to some extent contiguous in SnfR (see merged view, panel B) and that therefore this is a spurious correlation. If you don’t manually discard neighbouring ROIs, this is what you get. We added a note to clarify in the figure legend.

The idea that we select for stronger synapses is probably correct, even if we do not agree with the argument based on sensor affinities for glutamate (which are essentially the same). We added a note to mention this at line 617. But weak synapses contribute little to electrophysiological responses, and possibly little to physiological responses in general. iGluSnFR also undoubtedly reports extrasynaptic glutamate too (the responses are too broad for anything else).

Our correlation between depression in electrophysiology and fluorescence for SnFR-gamma2 is exceptionally good. For iGluSnFR it is non-existent. We think this is an argument that targeting selects the most meaningful signals in a physiological sense, and that we have demonstrated this.

3) Quantal analysis: As the biological meaning of the quantal parameters is best defined in single synapses, I will first comment on the single synapse experiments (Figure 8, Figure S6, Figure S8). The authors analyze single ROIs at three different calcium concentrations, resulting in increasing response amplitudes. This is interpreted as multi-vesicular release from a single synapse. While I follow this general interpretation, it seems the sensitivity of sensor/imaging system is not sufficient to detect the release of individual vesicles. In 0.5 mM ca^2+^, the recorded SnFR-y8 traces show no difference between simulated and non-stimulated epochs, strongly suggesting that the fluctuations in fluorescence intensity are noise. Consequently, the histograms of 0.5 mM ca^2+^ responses show no separation between failures and sucesses, just a single peak. To assume that this peak corresponds to the release of a single vesicle, the quantal response (q), is wrong. The example traces from iGluSnFR at 0.5 mM ca^2+^ look slightly more promising (Figure S6), but again, the histogram does not show a separation between failures and successes, and the compound histogram treats the entire first peak as the quantal response (i.e., assumes there were no failures in 0.5 mM ca^2+^). A method that lacks the sensitivity to detect single vesicle fusion is not useful for quantal analysis: Due to increasing variance and indicator saturation, the separation between multi-vesicular events will be less clear, not better, than the separation between uni-vesicular events and failures.

We have been quite open about limits of the quantal analysis in the paper. The major point is that a targeted sensor is better for quantal analysis, because many more ROIs are usable (can be fit) and we very clearly see regularly-spaced dips in, for example, the 2 mM histogram for SnFR-g2, but not for regular SnFR. This difference matches well previous work where spatially confined SnFR could give histograms with dips (Jensen, T. P., et al. 2019. Nat Commun *10*, 1414; Duerst, C., et al. 2020. Biorxiv), but unconstrained SnFR could not (Soares, C., et al. 2019. Front Synaptic Neurosci *11*, 22).

What we want to show here is that we can estimate N and Pr from a comparatively simple set of measurements, only 150 stimuli, because the SnFR-g2 construct is stable, unlike responses from iGluSnFR. We already mentioned in the discussion that it is hard to resolve the release of individual vesicles with SnFR-g2 at 0.5 mM (the reviewer mentions SnFR-g8, we do not show this, perhaps SnFR-g2 is intended?). We now reiterated this point at the end of the results (line 477), However, global fitting gives us an excellent estimate of quantal size. In this global fit, we found that omitting the failure peak helps to get a more robust fit. We are fitting 20+ ROIs in series, semi-automatically.

We understand the reviewer’s concern, could not the first peak just be noise? Please look at the pooled histogram from SnFR-g2 (now Figure 9D). There are three perfectly spaced peaks (multiples of 0.06), with a dip after each one. In fact, we get an excellent estimate of quantal size (which we use for the rest of the fits) from the spacing of these peaks. This is not consistent with the first peak being noise from overzealous fitting. These are typical data from a single ROI. We added text to describe this aspect better (after line 875 in the results).

Of course, the reviewer is right that detection of multivesicular events would be hurt by saturation and increasing variance, but for our sensor this appears only to happen after the 3^rd^ peak or so. We used the summed histogram to estimate the peak spacing. Our fit for individual calcium concentrations assumes that there is a failure peak at zero. Our method of peak detection does not give negative numbers, meaning we cannot plot a true “failure” peak (symmetrical around zero) as the reviewer would like. But we estimate failures very well with 0.5 mM glutamate – they are in the first bin of the histogram, they are fitted, and we have a *P*_r_ of about 20% across the 6 cells we include for this condition. We did initially try to do peak detection including negative values from failures, but the results were much less reliable, so we settled on this approach to give more reliable output from our detection routine (we give details around line 875). However, we will revisit this point and endeavour to obtain a more convincing “failure peak” in future work.

It seems like new versions of iGluSnFR using the principle that we established here will be brighter, and this will combine favourably to give a really nice quantal response. We will explore modifying our software accordingly but this is beyond the scope of this work.

The difference between stimulated and non-stimulated epochs is not pretty, we admit, but there is a difference. We already showed this in detail in the supplementary material (now Figure S7).

4) Quantal analysis of autaptic connections (Figure 7): The authors use the CV method developed for paired recordings, which seems appropriate for an autaptic neuron. Compared to the analysis of single synapses, the meaning of N is less well defined here. Early studies assumed that one synapse can only release one vesicle at a time, which makes N the number of connecting synapses and Pr the synaptic release probability. For autaptic cultures, multi-vesicular release is established, so N is the number of "release sites", several of which can be located in a single synapse. Thus, N must be larger than the number of recorded synapses, which in this case is the number of ROIs.

We apologise that the answers to these concerns were not made clear enough in the original submission. If we follow the logic outlined above, it is important to note that we are looking at the coefficient of variation at individual ROIs. Therefore, *N* is either the number of release sites in an ROI, or more generally the number of release sites multiplied by the number of synapses in the ROI.

We compare single ROI data to literature data on single connections from electrophysiology- in our opinion this is quite meaningful.

We now explain this point around line 393, and commit to explicit definitions of *N* and *P*_r_.

It is very difficult to understand Figure 7. Trains of 5 AP are evoked at 5 Hz ten times in 3 different Ca concentrations and recorded in an unspecified number of ROIs (synapses). In panel c, facilitation/depression during the train is compared for electrical and optical recordings. I assume these points (n=6) correspond to different cells, not different ROIs (please indicate the point corresponding to the cell analyzed in panels a and b). SnFR-y2 responses are better correlated with the electrical responses in 2 and 4 Ca, but SnFR-y2 synapses tend towards depression compared to iGluSnFR synapses. This population difference in EPSC dynamics raises the nasty possibility of presynaptic effects caused by SnFR-y2 expression (but due to low sample size, may just be a fluke).

We apologise that Figure 7 (now Figure 8) was hard to understand and led the referee astray. In panel C, we show the results from 6 cells for each condition. SnFR-g2 shows an indistinguishable range of depression/potentiation in electrophysiology from iGluSnFR (range of data on y-axis for each condition for iGluSnFR and SnFR-g2 respectively: at 2mM ca^2+^ : 0.8-1.1 and 0.7-1.25, and at 4 mM ca^2+^ 0.58-0.8, 0.53-0.78). It’s a small sample size but there is no systematic “nasty” effect of the sensor.

On the other hand, the correlation between the extent of depression in electrophysiology and the automatically-selected ROIs is stunning (but for SnFR-g2 only).

We now indicate the cell analysed in panels a and b with a filled symbol and note this in the legend. We now summarise in each figure panel what the experiment is with an explanatory title.

In Figure 7D, mean and standard deviations are calculated, but the basis is not clear. I am assuming each ROI produces 5 points in panel D, each averaged over the 10 repeats. It would be helpful to explain this more clearly. The CVs (slopes in panel D) are related to N and Pr. As the authors state, they end up with 3 numbers for an equation system with 4 variables. It is not clear to me how fitting can help in this situation (as there should be no error, regardless which number is chosen for N). So perhaps panel F shows the numbers they chose for N, perhaps based on the number of ROIs? (The legend of F is cryptic. Please indicate in E and F which points correspond to the cell shown in D.) This would be based on the assumption of univesicular synapses, though, which they refute in Figure 8. In any case, they point out that this N is similar to the N reported by Bekkers and Stevens, who were interested in the number of release sites connecting two neurons. The numbers (N = 4-7) seem very low for a connection producing 2 nA EPSCs (400 pA per uniquantal synapse?). It could of course be argued that the autaptic connection is made of many more synapses than the few that are sampled optically. The N determined by optical quantal analysis, however, has to be higher than the number of ROIs (active synapses), it cannot be lower. If the N is assumed to be equal to the number of ROIs (uniquantal synapses), it makes no sense to call the process quantal analysis (as in this case, the distribution of N values (Figure 7F) is not the result of any fitting procedure). Without understanding the basis of N, I cannot interpret the meaning of Pr (synaptic? vesicular?).

Apologies, we could have again explained much better here in order not to leave the reader confused. First of all, the autapse has many connections to produce nA synaptic responses, but we are looking at individual ROIs and as explained above, the *N* value refers to each ROI. We took the mean and SD from individual ROIs, for each pulse in the train (average and SD of the 10 first pulses (one from each train of 5), average and SD of the 10 second pulses, and so on). These averages and SDs were similar across the 5 pulses in the trains so we averaged them and plotted these points. Therefore, each point represents the mean and SD of a single ROI during the 10 x 5 pulse train stimulation. We could have separated out the responses according to their order in the train, but it seemed excessive, particularly given the limitations of this CV method.

Therefore the *N,* as we explained in the text, is related to the number of sites per ROI. We think this relates well to the original (and common) meaning from Katz: the probability of vesicular release from a synapse. This is a global analysis, we could have calculated for each ROI (synapse) because we have these ratios. But as we noted, the fit is not well defined for *N*. It means overall that the approach is not particularly satisfactory (and we would not recommend it, we now state this explicitly in the text around line 454).

We are grateful to the referee for asking the question as to how N is constrained. We went back an investigated this point again, which deserved more attention. What is really clear is that *N* < 3 does not work– the associated *P*r values are nonsensically saturated at a high *P*. We now show this in a new panel (Figure 8E), and took an alternative approach. We now took the lowest value of N which gave Pr < 50% at 0.5 mM Calcium. There might be an interesting connection here to the optical method selecting for stronger synapses – the *P*r values from CV analysis are generally too high compared to electrophysiological estimates which have been studied in much more detail.

We now indicate in panels E, F and G and in the legend which points and lines correspond to the relevant cells in D (with solid line and solid symbols). We now try to explain better (around line 400) and added notes to the text to point out why we think that we observe multivesicular release optically.

Suggested improvements:Provide conclusive evidence for successful synaptic targeting, e.g. by homer/y2 co-transfection of individual neurons and high-resolution imaging. Targeting is expected to be most specific at low expression levels.

We appreciate this suggestion. We now live labelled SnFR and variants with antibodies, and compare with PSD labelling in fixed cells. Labelling SnFR variants allowed us to see the mild enrichment of our targeted sensors at synapses (new Figure 3).

I am not sure how to improve the quantal analysis (more excitation light?). A symmetrical failure peak should appear exactly at df/f = 0 which is absent from all histograms. Perhaps the problem is background subtraction (leading to division by zero) or the handling of negative df/f values. If there is no way to separate failures from low probability release events in individual trials, I will remain skeptical about multi-peaked compound histograms apparently separating 4 from 3 simultaneously released vesicles.

The symmetrical failure peak is not seen due to the way in which we have done the analysis. The referee is right, we do not take negative values. We now discuss this explicitly in the methods (starting at line 878). Failures are simply in the first bin (starting at zero), and are accounted for in the fits. We are also skeptical about the high order events (like 3 from 4) and now note explicitly in the text that the peaks above 3 or 4 components are washed out (around line 450).

Please commit to a clear definition / interpretation of the extracted parameters from the coefficient of variance method, specifically in relation to the number of ROIs. At present, the biological interpretation of N and Pr with respect to autaptic synapses is unclear, at least to me.

As outlined above, we rewrote the text here. We commit to clear definitions of N and Pr (around line 392), and emphasise that this CV approach is not ideal – we think the quantal analysis of histograms for individual ROIs is better.

Reviewer #2 (Recommendations for the authors):1. According to the authors, one principal advantage of their approach is that SnFR-y2/8 provides a more 'spatially precise' signal compared with iGluSnFR. Clearly, an optical sensor that is proposed as expressed within a nanoscopic membrane domain will yield a more spatially constrained signal compared with the sensor expressed evenly over cell membranes. However, this simply reflects the sensor distribution properties rather than anything else. In words, SnFR-y2/8 will tend to report the brightest glutamate signal where SnFR-y2/8 is accumulated, rather than where the glutamate concentration is highest (e.g., release site proximity). In contrast, the sensor homogenously distributed in space, such as the original iGluSnFR variants, should provide unbiased readout of glutamate hotspots. It appears therefore that the authors strategy is somewhat self-defeating.

This comment to some extent represents the canonical view in the field, which we felt with this work we convincingly rebut. The reviewer is entitled to their opinion, but there are some observations in our work and published work that we believe speak against it. Most importantly, the team at Janelia have adopted exactly our approach (targeting with the Stg-C-terminus) for their new best in class indicator.

A. We show that iGluSnFR ROIs from postsynaptic sites (detected in an unbiased way) are massive. They cannot correspond to synapses. Perhaps this is only in the autapse system but we think it’s why most successful quantitative work to date was done in the Schaffer Collateral terminal. A non-targeted sensor is limited in terms of what you can look at. In contrast, the principle underlying our sensor should allow less bias in the choice of neuron, type of neuron, or its geometry.

B. Synapses can be rather densely packed on the dendrite (>1 per micron). In this case, regular SnFR is not useful, the signals from neighbouring synapses will overlap. Particularly, the dominant signal from SnFR could be spillover, whenever release occurs reasonably frequently. This effect might also cause saturation of the (high-affinity) sensor, compressing responses in an unpredictable way.

C. If the referee’s contention that iGluSnFR should be used as an unbiased reporter of glutamate hotspots were correct, one might expect published, quantitative work expressing iGluSnFR at postsynaptic sites. This is not the case. Rather, almost all quantitative work to date used presynaptic terminals to get spatial restriction to individual sites. In contrast, one paper using CA1 pyramidal cell dendrites needed extensive mathematical modelling to approach quantitative insight (and did not see dips in the quantal histogram, Soares, C., et al. (2019). Front Synaptic Neurosci *11*, 22.).

D. The idea that PSD ligands might accumulate at a non-trivial distance from release sites is refuted by published work (for example Li, T. P., and Blanpied, T. A. (2016). Front Synaptic Neurosci *8*, 19.) Particularly, the PSD and the release sites are aligned on the nanoscale. We now include new data on colocalization to show the better overlap at PSD sites – the new Janelia preprint shows similar data.

E. We also show that a non-targeted sensor can report spurious signals that should be discarded. Unfortunately, spatial bias is not the only possible source of bias.

Overall, our sensor certainly has some limitations. This being said, we feel we have given enough evidence to indicate that the approach is not self-defeating. In fact, it seems to be taken up as the way forward.

We now mention that a potential disadvantage of our sensor is that non-synaptic signals would be missed (around line 623).

2. The authors use TIRF imaging, and single-line diode lasers as an excitation source. This type of imaging is only suitable for monolayer cultures: whether their sensor will be efficient when imaged in organized brain tissue using two-photon excitation is not clear. The claim on a methodological advance appears therefore premature.

We did not use TIRF imaging. -we never mention it (the acronym is only in the name of the objective). We used diode lasers for some experiments and LED illumination in others. Regular SnFR has been used a lot in 2P imaging, and our sensor arguably has better signal to noise, and very little reason to believe the photophysics should be different. We didn’t change that part of the sensor at all.

What we do is to record an arbitrary number of inputs to a single neuron (an autapse in our case). This is our methodological advance. This experiment was not possible before.

3. The other key statement is that iGluSnFR is prone to photobleaching (rundown) more than is SnFR-y2/8. First, fluorophore's photobleaching properties in 1P as opposed to 2P mode could be very different, which has not been addressed or explored. Second, this observation is surprising because several recent studies have documented a fairly stable iGluSnFR signal over multiple cycles of glutamate release imaged at 'quantal' resolution, both in 1P and 2P excitation regimes, both in cultures and in acute slices (e.g., Tagliatti et al., 2020 PNAS 117: 3819; Jensen et al. 2019 Nat Commun 10: 1414). The authors do not seem familiar with these studies. Dye photobleaching depends on multiple imaging parameters starting with laser power: this has not been investigated consistently in the present work.

Because we do not know exactly what is happening to SnFR, we did not talk about photobleaching. Whatever happens occurs only at a subset of sites (about half of them), suggesting it is not due to illumination *per se*. Overall, the photophysical performance of our sensor seems quite similar to SnFR. The fluorophore is the same and we would expect 2P performance to be similar.

We now include a supplementary figure (new Figure S5) where we analyse the rundown and compare to the electrophysiological recording. Please see our response to referee 1 for more details. The results are interesting, we are grateful for the push to examine more closely. There is rundown of the synaptic currents, but iGluSnFR substantially overestimates it. Our sensor seems if anything to underestimate the rundown in current. We write about these new analyses near line 320

We know the papers suggested. We cited similar manuscripts; those mentioned by the referee above are now also cited for the same purposes (on lines 48 and 88, respectively). These are very good papers, but in the context of our work, we felt we might have to criticise their limitations, and we wanted to avoid this and be as neutral as possible. We apologise for being timid. These papers select individual examples of optical signals that work well for them. In the Tagliatti paper, they select broadly spaced presynaptic terminals by hand to localize SnFR. The Jensen paper is done on single identified connections on Schaffer Collateral, using presynaptic localisation

In our case, looking at a large number of inputs to a single neuron, this means going through movies, checking and selecting responses from hundreds of active ROIs by hand. This approach is neither scalable, nor reproducible. Our point is that, rather than collecting a mixture of gold and junk and manually selecting the gold, why not just collect good ROIs (our case)? This is essential if you want to examine arbitrary geometries, not just presynaptic terminals (for example, inputs to neurons in the brain). This approach is scalable to the number of synapses on a neuron (100-1000), rather than the number of Shaffer Collateral terminals at a connection (a single terminal).

4. Stargazin overexpression has been used as a principal tool for the iGluSnFR targeting to synapses, but the authors report that this renders a proportion of synapses nonresponsive to glutamate (Figure 4). That the method interferes with the physiological integrity of synaptic circuits, or at least requires some additional experiment-specific manipulations to minimize it, does not speak in its favor.

Point taken, but most experiments require “experiment-specific manipulations” to minimize problems in the data collection. We measured a problem (that few would bother to check) and we showed how we largely ameliorated it (through later infection), and learned something new in the process. Agreed, not ideal for our sensor, but valuable information, we feel.

5. The signal-to-noise ratio of the SnFR-y2 signal does not appear improved compared with that of iGluSnFR (Figure 8A).

In some contexts the SnFR-g2 SNR is higher (now Figure 4), in some not (now Figure 9). But high signal to noise is not everything, if some of the signals turn out to be unrelated to what you are trying to measure (see our now-Figure 8). Our work shows that the signal to noise of the targeted sensor is quite similar to the original SnFR, but the targeted sensor is in other ways a better reporter.

6. The quantal-analysis histograms presented here (Figure 8C-D) appear noisier hence less reliable than similar or related analyses in the aforementioned publications that employed iGluSnFR.

Again, this might be explained by selection and the type of experiment. The “nicer” histograms come from presynaptic expression, perhaps using hundreds of stimuli. We used only 50 stimuli per condition, so the individual histograms are not so beautiful. But overall the data are very rich, because we have 40 or so ROIs per field of view, and collect across three different conditions. The overall lack of bleaching of the SnFR-g2 sensor (now shown in supplementary Figure S5) suggests one could collect more data to have more reliable histograms.

Reviewer #3 (Recommendations for the authors):Hao and colleagues developed new variants of the glutamate sensor iGluSnFR, termed SnFR-γ2 and SnFR-γ8, that are fusion proteins with postsynaptic density (PSD) proteins Stargazin and γ-8. This chimeric protein thus localizes specifically at the PSD. These new variants are characterized, using heterologous expression systems and hippocampal neuron microisland cultures, allowing one to monitor autapses with simultaneous electrical and optical access. Overall, SnFR-γ2 outperforms traditional iGluSnFR in terms of signal localization; presumed single synapses are observed with limited "spillover" of signal to neighboring regions, and imaged transients are amenable to traditional noise analysis. Some concern regarding competition for PSD membrane is raised due to overexpression of these variants, but it appears that such overexpression artifacts can be avoided by ensuring that SnFR-γ2 is delivered after synapses are largely formed. This could be an important tool for the field, as it improves one's ability to resolve the activity of single glutamatergic synapses with sufficient signal to noise. Work here has shown feasibility in cultures systems. Future work will need to show similar performance levels in acute slice and in vivo, though based on past observations with iGluSnFR, this performance is likely within reach.

We thank the reviewer for their balanced critique.

The main question I have is one of extensibility: can this approach work in more intact systems, or even in vivo? I recognize that asking such a question involves an entirely new dataset, and am not proposing that the authors engage in such an effort after already doing an excellent job characterizing these GluSnFR variants. Rather, I'd hope that the authors would expand on their discussions, which is largely focused on questions of release dynamics observed with their sensors, to also include potential technical advantages or limitations of these variants in other preparations.

Thank you for this kind recognition. We added a line that better performance is needed in vivo, and that our sensors (also with reference to the new preprint from Janelia) are likely better for quantitative work in vitro (around line 599). We do not want to speculate further at this stage.

Comments below are aimed at improving interpretation of data reported herein:1) A control for infection is needed for autapse data. Please make parallel recordings in cells infected with control viruses that lack any glutamate sensor to determine if these currents are in the normal range at these ages.

Thank you for this suggestion. We added these data to the figure (now Figure 6C) – a separate set of recordings of autaptic neurons infected with a GFP AAV. The currents are a bit bigger in this condition.

2) Data in Figure 5C and F should be analyzed quantitatively by calculating correlation coefficients for each pair of data. If there are differences in the relative separation of each region (I understand that these were chosen by hand) then a potential comparison could be made by plotting correlation coefficients vs. centroid distance of paired ROIs.

We now provide the correlation coefficients for the data we show, which give a very conclusive answer along the lines that the referee imagined. Thanks for this great suggestion.

3) Data in Figure 7C, 0.5 mM. There is an obvious outlier near 7 P5/P1 for the SnFR-gamma2. This is likely due to a very low P1 value for this one cell, indicative of excess failures. I'd be curious to know if any correlation holds if this one datapoint is held out of the dataset. If so, perhaps the authors could explain this observation in the main text.

Thank you for this suggestion. Briefly, we looked to see if removing this data point gives a better correlation, but the correlation is still poor so in the interest of transparency, we left it in. We now explain this in the text (near line 373).

4) Figure 7, data related to optical quantal analysis. Considering that these are autaptic recordings, with superb electrical access, one should be able to perform traditional electrophysiological quantal analysis and determine whether SnFR is allowing for identification of all synapses. This is a critical analysis that should be made on a cell-by-cell basis, paired with optical analyses; however, if this is not feasible at this time, some information could be gleaned from separate recordings, given that you observed fairly consistent numbers of sites optically (4 to 7). This would address your concern that the detection methods bias towards high Pr synapses. Though, if there are more synapses made a significant distance from the coverslip, then they would never be imaged under TIRF microscopy, obviating this request.

Unfortunately not – the autapses have way too many synapses (possibly hundreds) for quantal analysis with electrophysiology. The *N* we derive is per ROI. Although we did not use TIRF microscopy, the reviewer is still correct, we cannot pick up all synapses because of the 3-D nature of the cell, and lack of focal depth with high NA objective. However, those that we do detect with SnFR-g2 offer a very good measure of the behaviour (depression) of the entire group as ascertained from electrophysiology (whereas iGluSnFR ones do not) – this is shown in the correlation in the now Figure 8C.

[Editors’ note: what follows is the authors’ response to the second round of review.]

The manuscript has been improved but there are some remaining issues that need to be addressed, as outlined below:This manuscript is an attempt towards synapse specificity for glutamate probes. The current probe's utility is limited to specific cases, and it has what could be considered advantages and drawbacks, even in that specific case (CMOS imaging of cultured cells). It is commendable how carefully the authors characterized side effects of this sensor on synaptic function. The reviewers agreed that several issues require clarifications in the text.

We agree with these points and we are grateful for the recognition of the care we have taken.

1. Please explain that any glutamate rises or waves in the tissue, any significant glutamate spillover signaling, or even fluctuations in tissue optical conditions, could be falsely perceived as local synaptic signals by an PSD-constrained sensor rather than by an evenly distributed sensor. This is a key limitation of the current method.

This is a good point and we have clarified it in the revised text (around line 662).

2. The authors seem to insist that the use of iGluSnFR or any sensor labeling is generally disadvantageous. It is advantageous in most cases. In terms of this sensor's fluorescence properties, there is no evidence for improved performance in the manuscript.

We have clarified that our sensor does not have improved fluorescence properties (around line 658), and already noted that untargeted sensors certainly still can have excellent performance, particularly if confinement can be achieved (line 87). We direct the readers to our recent review on this topic in JNS Methods for a wider discussion of this theme (near line 667),

3. Please consider the specific points raised by the reviewers and address them in the text to the extent possible.

Our point by point outline of the changes to the text follows below.

Reviewer #1 (Recommendations for the authors):The authors have provided their explanations and rebuttal regarding the previous comments, albeit without new experimental evidence.

We did actually provide some new experiments (Figure 3), showing the localisation of the sensor.

1. It appears that the authors did not fully understand the main objection. Or perhaps it was not explained clearly enough. To reiterate: SnFR-y2/8 expressed locally at the synapse may in fact sense and report glutamate that is released elsewhere in the vicinity, thus giving a false impression of its local synaptic release. In other words, SnFR-y2/8 may report the spillover signal as well as the actual synaptic signal.It is true that, in 2PE mode, in organized brain tissue, iGluSnFR will report optical signal integrated, due to diffraction of light, within the PSF depth of ~800 nm, etc., and thus may include glutamate release events, if any, from nearest synaptic neighbors. However, the same will happen if such neighbors express SnFR-y2/8: their optical signal will still be integrated within the diffraction-limited PSF in 2PE imaging mode.In the case when no glutamate spillover ever occurs in the preparation of interest, both iGluSnFR and SnFr-y2/8 will report true synaptic signals when such happen. In this case, however, glutamate rises or waves arising from other sources, such as astroglia, will still be reported by SnFr-y2/8 very locally, giving an impression of local synaptic events – whereas iGluSnFR will report the entire glutamate 'landscape'. Thus, SnFr-y2/8 does not seem to have any principal spatial-resolution advantage over iGluSnFR.

As suggested by the editor, we have added some clarifying text that our targeted sensor will still report glutamate from *all* sources. The examples given by the reviewer are excellent and we use them (around line 662). We moved the final sentence of the discussion forward to offer the potential solution: to have a multiplexed output (as suggested by reviewer 3 in the first round of reviews).

2. Authors' attention is drawn to yet another recent publication (Mendonca et al. 2022 Nat Commun 3497) showing an excellent stability, S/N ratio, etc. for highly localized glutamate release events detected with iGluSnFR. That this sensor is performing not as satisfactory in the present authors' hands cannot be a basis for their extrapolated claim.

We are aware of this work and are happy to cite it (in the introduction line 88 and around line 673). It’s a beautiful study measuring the outputs of a single neuron. It does not correspond directly with what we have done – we measured at confined, postsynaptic sites, in order to detect the inputs to a single neuron, a complementary approach. We congratulate the authors of the Mendonca paper on their work, it’s impressive. Some of the techniques they use certainly exceed the sophistication of our work.

As the referee probably knows, but for the sake of transparency, Mendonca and colleagues used a newer,brighter variant (SF-iGluSnFR) that was not available when we started our work. In the Mendonca paper, the sensor is not tethered, but importantly, the sensor is confined to the presynaptic neuron. This microanatomical restriction apparently helps with the signal localisation (the same effect was seen at the Schaffer terminals). We already noted that in some other work, physical confinement of the sensor was enough (around line 87). Presynaptic tethering did improve performance in other work that we cited (Kim et al., JNS 2020) although the Mendonca work suggests that this is not really necessary.

We do not understand these discrepancies, and we agree, our work is not definitive in this question. More work is needed. We have added a note in the text to this effect (line 673).

3. The authors concede that the S/N ratio of their probe readout is probably no better than that of iGluSnFR.

This point is now quite clear in the text, we hope (explicitly stated at line 658).

4. The authors have made no attempt to check their probe performance in 2PE mode and/ or in organized brain tissue.

We are sorry, we do not have the resources for that (yet). But we are grateful for the reminder about how important this is. We hope to address this point adequately with future work.

5. The amplitude histograms shown in Figure 9C-D do not match satisfactorily the best-fit quantal analysis curves. While the authors acknowledge the difficulty, the reasons for displaying unsatisfactory quantal analysis data are not clear.

We take the point. As reviewer 3 has commented, the separation of quantal peaks is really what determines the fit – to get a good agreement with the small amplitude components we might need to sample more data. We feel that the more important point is that we could semi-automatically obtain similar information from 40+ such sites in the same neuron at once. There remain only a few examples of such an analysis.

We added a note to the text (around line 505) to mention that the incomplete agreement between the fits and the histogram might be ameliorated by collecting more responses.

6. The entire concept would have made a much greater impact if the authors aimed to express the sensor at a specific, functionally or genetically distinct, sub-population of synapses.

We agree, this would be fantastic. We would like to point out, that the prevailing view (up to this preprint) was that targeting iGluSnFR was a very bad idea. Now we showed, and the Janelia preprint also shows, that targeting can be neutral, or even advantageous in terms of the signal collected. The greater impact that the referee seeks from molecular/cellular targeting? We seek it too and we hope now to aggressively pursue targeting in various contexts (as the reviewer describes). Maybe other investigators will too. We hope so.

Reviewer #2 (Recommendations for the authors):The authors have addressed concerns raised by all three reviewers to the best of their ability. Their commentary that untagged SnFR is likely detecting large hotspots of both synaptic and spillover is taken well. Though the concern regarding how well the tagged variant works remains in question. If one can only resolve a few synapses (less than 10 per cell were analyzed) of what the authors suggest are ~100 potential autapses, are these few ROIs representative of the whole? This may be a good point of discussion.

Thank you. We added a line of discussion (around line 530) about the disparity between the amplitude of the synaptic current and the number of ROIs that display activity. We know that this question has been bothering a few groups for years. Perhaps with further work we can resolve it.

Given the way in which eLife is modifying its review process and overall publishing criteria, this work should be accepted. It represents a new tool that is well characterized. Advantages and flaws are discussed well.

Thank you.

Reviewer #3 (Recommendations for the authors):In their revised version, the authors address questions about synaptic localization of their GEGI by demonstrating good overlap with the PSD95 signal (new Figure 3). The explanation of Figure 8 has been much improved, quantal parameters are now well defined. The correlation between electrical and optically measured depression during a train (Figure 8c) is indeed much better for their targeted indicator compared to iGluSnFR, which is a strong argument that synaptic (and not extrasynaptic) glutamate is reliably measured by their sensor. In this context, it is an advantage of the autaptic model that all synapses on a given neuron have the identical history of activity and therefore express similar (but cell-specific) short-term plasticity. I do not like the analysis of rundown in Figure 7F: Splitting a continuous distribution into 2 groups, using an arbitrary threshold, then counting the number of cases (ROIs) on either side of the threshold, is not good statistical practice. The analysis in Figure S5 I like much better, it shows no significant difference between the two indicators. As the authors cannot offer a convincing mechanistic explanation why a difference in rundown would be expected, I suggest downplaying this point (getting rid of Figure 7E-G, sticking with the message of S5).

We are grateful for these supportive comments and for the thoughtful analysis of the data. We agree that the arbitrary split is not ideal, but we did it in order to orient the reader with a concrete example of a difference between broadly usable and unusable data.

Aiming to follow the spirit of the referee’s suggestion, we demoted Figure 7E-G to supplementary material (Figure 7—figure supplement 1) and added a caveat that this approach is illustrative and not statistically rigorous (around line 315). We also provide a stronger statement that we have no mechanistic explanation for the rundown of individual ROIs (around line 571).

Apart from this quibble, the science is sound and well presented. The separation of quantal histogram peaks is impressive and certainly aided by localizing the indicator to the places of highest glutamate concentration. Extrasynaptic indicator molecules, exposed to a near-continuous range of glutamate concentrations, would be expected to widen the peaks considerably (as shown in the iGluSnFR example in Figure 9c). For future tool development and targeting efforts, the technical information and precise measurements are very useful, even as a somewhat cautionary tale with regard to potentially severe side effects of tool expression.

Thank you for these supportive comments.